# Label-free three-photon imaging of intact human cerebral organoids for tracking early events in brain development and deficits in Rett syndrome

**Murat Yildirim[1,2,3]\*, Chloe Delepine[1,3], Danielle Feldman[1,3], Vincent A Pham[1,3], Stephanie Chou[1,3], Jacque Ip[1,3,4], Alexi Nott[1,3], Li-Huei Tsai[1,3], Guo-Li Ming[5], Peter TC So[6], Mriganka Sur[1,3]\***

[1]Picower Institute of Learning and Memory, Massachusetts Institute of Technology, Cambridge, United States; [2]Department of Neuroscience, Cleveland Clinic Lerner Research Institute, Cleveland, United States; [3]Department of Brain and Cognitive Sciences, Massachusetts Institute of Technology, Cambridge, United States; [4]School of Biomedical Sciences, The Chinese University of Hong Kong, Hong Kong, China; [5]Department of Neuroscience and Mahoney Institute for Neurosciences, Institute for Regenerative Medicine, Perelman School of Medicine, University of Pennsylvania, Philadelphia, United States; [6]Deparment of Mechanical Engineering, Massachusetts Institute of Technology, Cambridge, United States

**\*For correspondence:**
muraty@mit.edu (MY);
msur@mit.edu (MS)

**Competing interest:** The authors declare that no competing interests exist.

**Abstract** Human cerebral organoids are unique in their development of progenitor-rich zones akin to ventricular zones from which neuronal progenitors differentiate and migrate radially. Analyses of cerebral organoids thus far have been performed in sectioned tissue or in superficial layers due to their high scattering properties. Here, we demonstrate label-free three-photon imaging of whole, uncleared intact organoids (~2 mm depth) to assess early events of early human brain development. Optimizing a custom-made three-photon microscope to image intact cerebral organoids generated from Rett Syndrome patients, we show defects in the ventricular zone volumetric structure of mutant organoids compared to isogenic control organoids. Long-term imaging live organoids reveals that shorter migration distances and slower migration speeds of mutant radially migrating neurons are associated with more tortuous trajectories. Our label-free imaging system constitutes a particularly useful platform for tracking normal and abnormal development in individual organoids, as well as for screening therapeutic molecules via intact organoid imaging.

## Editor's evaluation

This manuscript will be of interest to stem cell and developmental biologists who aim to use newly emerging brain organoid models to understand the structure and function of the developing human brain. It presents a technological advance in imaging and describes an innovative method for labeling and tracking cells within organoids to enable the assessment of dynamic processes within the intact organoid. The method is validated in a disease model and addresses a challenge in the field of human stem cell modeling of assessing cells within the 3D structure.

## Introduction

Human cerebral organoids derived from embryonic or induced pluripotent stem cells are unique in their ability to recapitulate early events of embryonic brain development (*Lancaster et al., 2013*). These spheroid structures contain progenitor-rich zones around ventricle-like cavities akin to ventricular zones (VZ) from which neuronal progenitors migrate radially to generate the cortical plate (CP) (*Lancaster et al., 2013*; *Qian et al., 2016*; *Paşca et al., 2015*; *Mariani et al., 2012*; *Kadoshima et al., 2013*). Human cerebral organoids also recapitulate gene expression programs of the fetal cortex (*Qian et al., 2016*; *Velasco et al., 2019*; *Quadrato et al., 2017*; *Pollen et al., 2019*; *Luo et al., 2016*; *Camp et al., 2015*) as well as the fetal brain epigenome (*Luo et al., 2016*). Thus, they have been used to model neurogenesis-relevant human pathologies such as microcephaly (*Lancaster et al., 2013*; *Zhang et al., 2019*), licencephaly (*Bershteyn et al., 2017*), heterotopia (*Klaus et al., 2019*), Zika virus infection (*Qian et al., 2016*; *Garcez et al., 2016*; *Dang et al., 2016*), and idiopathic autism (*Mariani et al., 2015*). Rett Syndrome (RTT) is an X-linked neurodevelopmental disorder caused by mutations in the gene encoding Methyl CpG binding protein 2 (MeCP2). MeCP2 is a pleiotropic regulator of gene expression and impacts multiple components of brain development and function (*Ip et al., 2018*). Structural deficits described in postmortem RTT human brains include reduced cortical thickness, cell size and dendritic arborization (*Armstrong et al., 1995*; *Bauman et al., 1995*) and reduced cerebral volume in MR imaging of RTT patients (*Carter et al., 2008*). These deficits are paralleled by reductions in dendritic arborization, soma size and spine density described in RTT mouse models (*Fukuda et al., 2005*; *Kishi and Macklis, 2004*; *Shahbazian et al., 2002*; *Smrt et al., 2007*). 2D human stem cell models of RTT, generated by reprogramming of patient cells or genome editing, have revealed deficits in human RTT neurons including aberrant transcription (*Lyst and Bird, 2015*; *Li et al., 2013*; *Chen et al., 2013*; *Gomes et al., 2020*; *Trujillo et al., 2021*), impaired neuronal maturation and electrophysiological function (*Li et al., 2013*; *Tang et al., 2016*; *Kim et al., 2011*; *Farra et al., 2012*), and up- or down-regulation of key signaling pathways and activity-related genes (*Li et al., 2013*). Using fixed tissue slices from 3D cerebral organoids, we have recently demonstrated impaired proliferation of the progenitor pool and delayed maturation and presumed migration of neurons (*Mellios et al., 2011*), consistent with human postmortem deficits. However, the use of fixed tissues did not allow cell tracking and analysis of parameters such as speed and trajectory of neuronal displacement; thus the dynamics and cellular mechanisms of the presumptive migration deficit remain to be characterized. Significantly, live cell time-lapse imaging of radial migration of MECP2-deficient neurons from the VZ to CP inside intact live 3D organoids has not been performed so far.

The majority of analyses of cerebral organoids have been performed in sectioned tissue (*Bershteyn et al., 2017*; *Klaus et al., 2019*; *Li et al., 2011*; *Lancaster and Knoblich, 2014*; *Andersen et al., 2020*; *Bagley et al., 2017*; *Karzbrun et al., 2018*; *Lancaster et al., 2017*; *Miura et al., 2020*), although recent progress has been made in studying migration using labeled neurons in organoid models (*Klaus et al., 2019*; *Bagley et al., 2017*; *Birey et al., 2017*; *Birey et al., 2022*; *Xiang et al., 2017*; *Bajaj et al., 2021*), including assembloid preparations (*Birey et al., 2017*; *Birey et al., 2022*; *Xiang et al., 2017*). However, labeled live cell imaging generally requires phototoxic dyes and limited incubation conditions, and results as well in labeling mostly peripheral neurons and cells which are not ideal for long-term imaging and for finding ventricular like zones in 3D intact organoids. Even with viral labeling, subtle damage to neuronal integrity can be sometimes detected (*Yildirim et al., 2019*). Optically, cerebral organoids appear opaque due to the optical density of neuronal tissues and the large number of apoptotic cells in the center of 3D intact conditions (*Lancaster and Knoblich, 2014*). Thus, label-free and high-resolution deep-tissue imaging are ideally required to perform intact organoid imaging.

Third-harmonic generation (THG) is an intrinsic signal which results from tripling of the frequency of the excitation wavelength. THG signal is either generated at structural interfaces such as local transitions of the refractive index or inside the materials whose third order nonlinear susceptibility $\chi^3$ is higher. Thus, THG excitation of biological tissues occurs predominantly at interfaces that are formed between aqueous interstitial fluids and lipid-rich structures, such as cellular membrane (*Rehberg et al., 2011*) and lipid bodies (*Débarre et al., 2006*). Both standard three-photon fluorescence microscopy and THG microscopy rely on three-photon interaction between ultrashort pulses and tissues. Three-photon fluorescence microscopy and THG microscopy has been recently used to perform structural and functional brain imaging in anesthetized and awake mice (*Yildirim et al., 2019*; *Ouzounov et al.,*

*2017*). These studies are based on utilizing a green (GCaMP) genetically engineered calcium indicator (exogenous fluorophores) with their excitation wavelength (1300 nm) which provides peak absorption cross-sections for this indicator. In these three-photon fluorescence microscopy studies, three photons with enough peak power at these excitation wavelengths excite electrons from ground state to excitation states of this indicator. Then, these electrons release their energy while they return to their ground states. During this relaxation, they release emitted photons which have a large $1/e^2$ bandwidth of fluorescence emission (70 nm for GCaMP6s/f). Therefore, an exogenous label is required to perform three-photon fluorescence structural or functional brain imaging. On the other hand, THG microscopy does not need any label for structural three-photon brain imaging. In addition, three photons with enough peak power at the excitation wavelengths excite electrons from ground state to virtual state so that these electrons do not lose any energy when they come back to the ground state. Therefore, they release emitted photons which have a small $1/e^2$ bandwidth of emission (~20 nm). Therefore, the emission spectrum of THG imaging occurs at exactly 1/3 of the excitation wavelength.

Overall, three-photon fluorescence microscopy is valuable for performing structural and functional brain imaging with fluorescent dyes for which the excitation wavelengths are limited by the peak absorption cross-sections of these fluorophores. However, three-photon fluorescence imaging is prone to phototoxicity and photobleaching particularly with high peak intensity pulses (*Yildirim et al., 2019*). In contrast, THG microscopy provides label-free structural brain imaging (*Yildirim et al., 2020*) without problems of phototoxicity and photobleaching - which are its strengths for long-term live cell imaging of cerebral organoids. Optimized designs using high power lasers and high sensitivity photomultipliers have enabled the application of THG microscopy to 3D tissue microscopy by improving the imaging depth in tissues with varying scattering coefficients (*Débarre et al., 2006*; *Yildirim et al., 2015*; *Yelin and Silberberg, 1999*; *Gualda et al., 2008*). THG microscopy has been applied recently to the non-invasive monitoring of human adipose tissue (*Chang et al., 2013*), cell nuclei and cytoplasm in liver tissue (*Lin et al., 2014*) and subcortical structures within an intact mouse brain (*Yildirim et al., 2019*; *Ouzounov et al., 2017*; *Yildirim et al., 2020*). THG imaging provides the advantages of micron scale resolution, label-free imaging, deep tissue penetration depth, and non-destructiveness.

Here, we describe a custom-made three-photon microscope with optimized laser and optics design to perform label-free THG imaging of intact organoids, and its use in imaging organoids generated from RTT and control patients to assess early events of early brain development. We demonstrate that high resolution label-free THG signal can be collected from intact cerebral organoids. Comparing organoids from RTT patients and isogenic controls, we show that the ventricular zone in mutant organoids has larger volume, larger surface area, and lower ventricular thickness compared to isogenic control organoids. Finally, label-free live cell imaging provides a unique way of observing neuronal migration inside the VZ/SVZ and CP of developing organoids without any phototoxicitiy or photobleaching, and reveals that the smaller ventricular thickness in mutant organoids is associated with shorter migration distances, more tortuous trajectories, and slower migration speeds of radially migrating neurons.

## Results

### Optimized system for label-free THG imaging of intact cerebral organoids

We developed a three-photon microscope (*Yildirim et al., 2019*) and modified it (*Figure 1A–C*, see Materials and methods) to perform label free THG imaging of intact, uncleared, fixed and live organoids. Since low average laser power is essential for live cell imaging, we focused on optimizing laser and microscopy parameters to reduce the average power requirement for intact organoid imaging. First, we utilized moderate repetition rate (<1 MHz) to maximize the energy per pulse and reduce the average laser power requirement for THG imaging. Second, we minimized the pulse width on the sample by building an external pulse compressor. The pulse width on the sample was reduced to 27 fs in the deepest part of the organoids (see Materials and methods) enabling us to further reduce the average power requirement by two- to threefold compared to other three-photon studies in the literature (*Yildirim et al., 2019*; *Ouzounov et al., 2017*; *Weisenburger et al., 2019*; *Figure 1—figure supplements 1 and 2*). Finally, we designed all intermediate optics in the excitation and emission path to maximize the generation and collection efficiency of THG signal (*Figure 1—figure supplement*

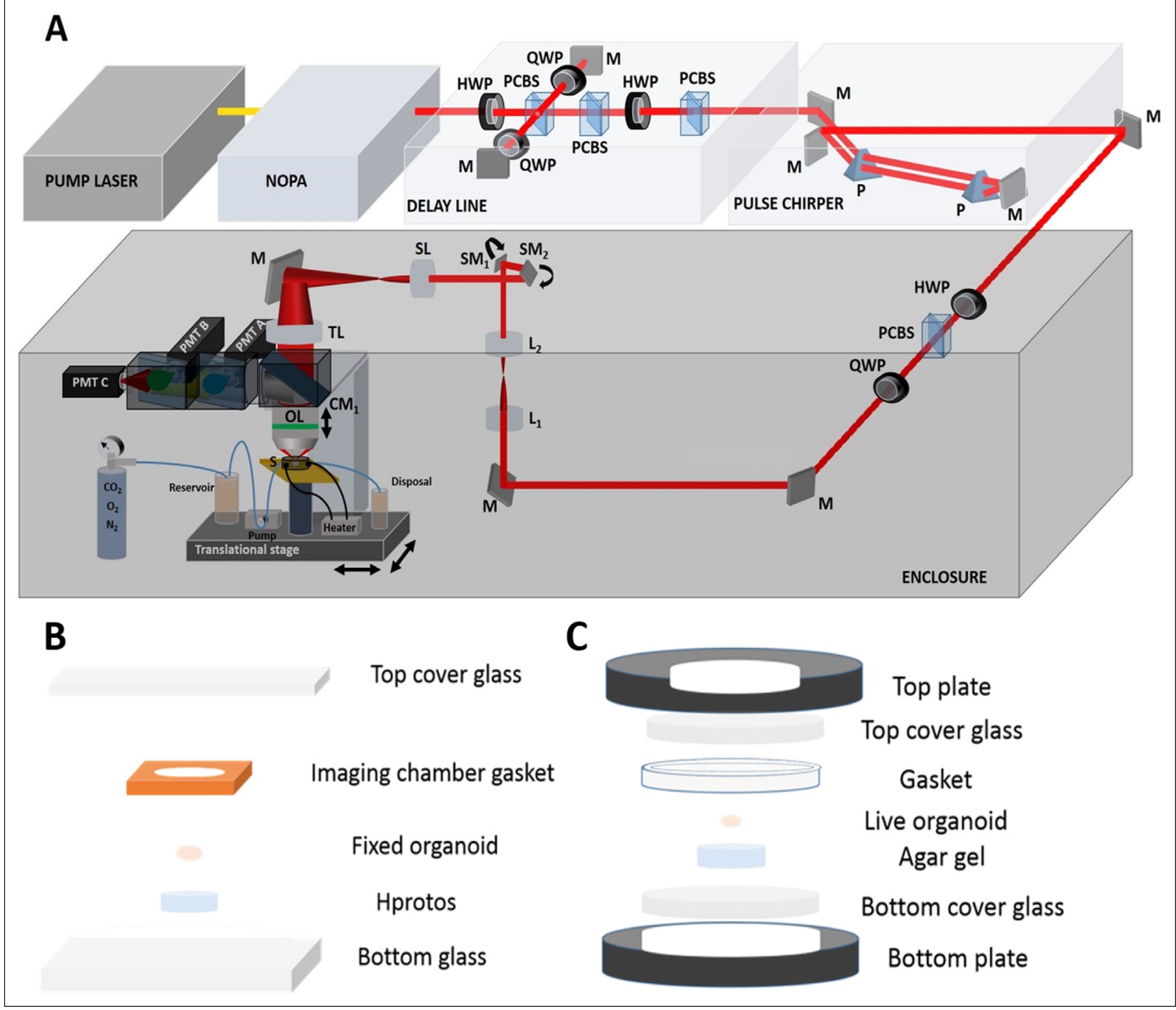

**Figure 1.** Three-photon microscope and imaging system. (**A**) Femtosecond laser pulses from a pump laser (1045 nm) were pumped through a noncollinear optical parametric amplifier (NOPA) to obtain 1300 nm excitation wavelength. Power control of these laser pulses was performed using a combination of a half-wave plate (HWP) and polarizing cube beam splitter (PCBS). A quarter-wave plate (QWP) was used to control the polarization state of the laser pulses to maximize the third harmonic generation (THG) signal. Laser beams were scanned by a pair of galvanometric scanning mirrors (SM), and passed through a scan lens (SL) and a tube lens (TL) on the back aperture of a 1.05 NA, 25×objective. The sample (S) was placed on a two-axis motorized stage, while the objective (OL) was placed on a one-axis motorized stage for nonlinear imaging. Emitted light was collected by a dichroic mirror (CM1), collection optics (CO), laser blocking filters (BF), and nonlinear imaging filters (F) and corresponding collection optics (COA, COB, and COC) for each photomultiplier tube (PMT A, PMT B, and PMT C). (**B**) Fixed intact organoids were placed inside imaging chamber gaskets on top of the bottom glass. Gaskets were filled with Hprotos and covered with a cover glass. (**C**) Live organoids were placed on top of the bottom plate of the incubator. Imaging gaskets were placed to secure organoids and cell medium was applied to fill the gaskets. Top plate was placed on top of the gaskets and the microincubator closed with 6 screws. Cell medium mixed with 5% $CO_2$, 5%$O_2$, and balanced $N_2$ was pumped to the chamber; incubator temperature was set to 37 °C.

The online version of this article includes the following figure supplement(s) for figure 1:

**Figure supplement 1.** Measurement of the pulse width of the laser at 1300 nm wavelength after it exits from the live cerebral organoid.

**Figure supplement 2.** Comparison of the effect of different pulse widths on the THG signal acquired in a cerebral organoid.

*Figure 1 continued on next page*

Figure 1 continued

**Figure supplement 3.** Comparison of one-inch and two-inch size collection optics by placing an iris (I) in front of the collection side of the microscope.

**Figure supplement 4.** In vivo point spread function (PSF) utilizing green retrobeads in anesthetized mice.

*3*). This optical design procedure enabled us to reduce the spherical aberration in the system and increase the collected signal by twofold compared to off-the shelf optics (*Ouzounov et al., 2017*; *Figure 1—figure supplement 4*).

## THG signals from intact cerebral organoids reveal distinct zones

Cerebral organoids were derived from two pairs of isogenic iPSC clonal lines (see Materials and methods). One pair (Line 1) was produced as previously described (*Mellios et al., 2011*) from one individual with RTT harboring a heterozygous single nucleotide deletion (frameshift 705delG) in the transcriptional repression domain of MeCP2. Because of the monoallelic expression of X-chromosome genes and clonal selection of iPSCs, the 'RTT-WT' line expressed exclusively the wild-type allele of *MECP2*, while the "RTT-MT" line expressed exclusively the mutated allele. The second pair of isogenic iPSC clonal lines (Line 2), produced as previously described (*Nott et al., 2016*), was from another individual with RTT harboring a heterozygous single nucleotide mutation *MECP2*R306C located in the transcriptional repression domain (TRD) or more specifically NCoR/SMRT interaction domain (NID) of MeCP2, which selectively blocks its interaction with the NCoR/SMRT complex (*Lyst and Bird, 2015*; *Lyst et al., 2013*). Because the *MECP2*R306C-derived iPSCs had skewed X-inactivation, and expressed only the *MECP2*R306C allele, the *MECP2*R306C RTT-WT line was generated from the MECP2R306C iPSCs using CRISPR/Cas9-mediated gene editing to correct the R306C mutation (*Nott et al., 2016*).

First, we established that cortex-like substructures could be distinguished in the organoids by cell density differences. In accordance with previous studies (*Lancaster et al., 2013*; *Qian et al., 2016*; *Velasco et al., 2019*), after 35 days of culture, the cerebral organoids demonstrated formation of well-organized structures composed of a ventricular zone (VZ)-like layer (KI67-positive proliferative cells) around a ventricle-like cavity and a surrounding cortical plate (CP)-like structure (Tuj1 or DCX-positive neurons) (*Figure 2A–B*, see details of the generation of organoids in Materials and methods). Notably, CP and VZ substructures demonstrated distinct cell density and morphology as shown by nuclear (7AAD) stain and plasma membrane specific dye (WGA) (*Figure 2C*). The VZ region, densely populated by neural progenitor cells and few immature neurons, could be markedly distinguished from the CP substructure with a less dense population of neurons. Thus, we subsequently used the cell density to localize the CP structures in three-photon fluorescence/THG imaging experiments.

Next, we demonstrated that high-resolution label-free THG signal could be collected from intact cerebral organoids. Nuclear stain was used as a control to localize the high cell density VZ from the low cell density CP (*Figure 2D*). Strikingly, THG imaging provided a clean delineation between VZ and CP as THG signal intensity was lower in the former area than the latter (*Figure 2C–D*). The intrinsic THG signal mostly arose distinctly from cell membranes. Consequently, the THG signal nicely revealed the typical radially elongated and bipolar radial glia cell morphology as well as the round-shaped somata of neurons (*Figure 2D–E*). Additionally, neurons in the CP and newborn neurons inside the VZ showed high intensity somatic signals (*Figure 2D–E*). Finally, we performed THG imaging of an intact organoid and then stained it with neuronal marker DCX. We were able to find the same field of view in our stained organoids and compare it with our THG imaging results (*Figure 2—figure supplement 1*). Our results confirmed that strong THG signal is mostly detected in the CP (DCX-rich, low cell density), rather than the VZ (DCX-negative, high cell density) (*Figure 2—figure supplement 1*).

To further validate that the intrinsic THG signal arose from neurons, we expressed green fluorescent protein (GFP) in control organoids by electroporation (see Materials and methods). We previously demonstrated that >90% of the GFP-positive cells in electroporated organoids are neurons (*Delepine et al., 2021*). Three-dimensional imaging of cells in organoids (~2 mm thickness) revealed that both WT and MT organoids in fixed and live conditions had >85% overlap between GFP and THG signals (*Figure 2—figure supplements 2–5*, see Methods).

Three-dimensional rendering of a~450-image z-stack with 2 µm increments revealed our system's ability to perform high-resolution imaging (*Figure 3—videos 1–2*; *Figure 3A–B*, left). We performed depth-resolved imaging of a 250 × 250 × 900 µm³ region selected to include a cortex-like structure, centered around the ventricle-like cavity, for both RTT-WT (*Figure 3A*) and RTT-MT organoids

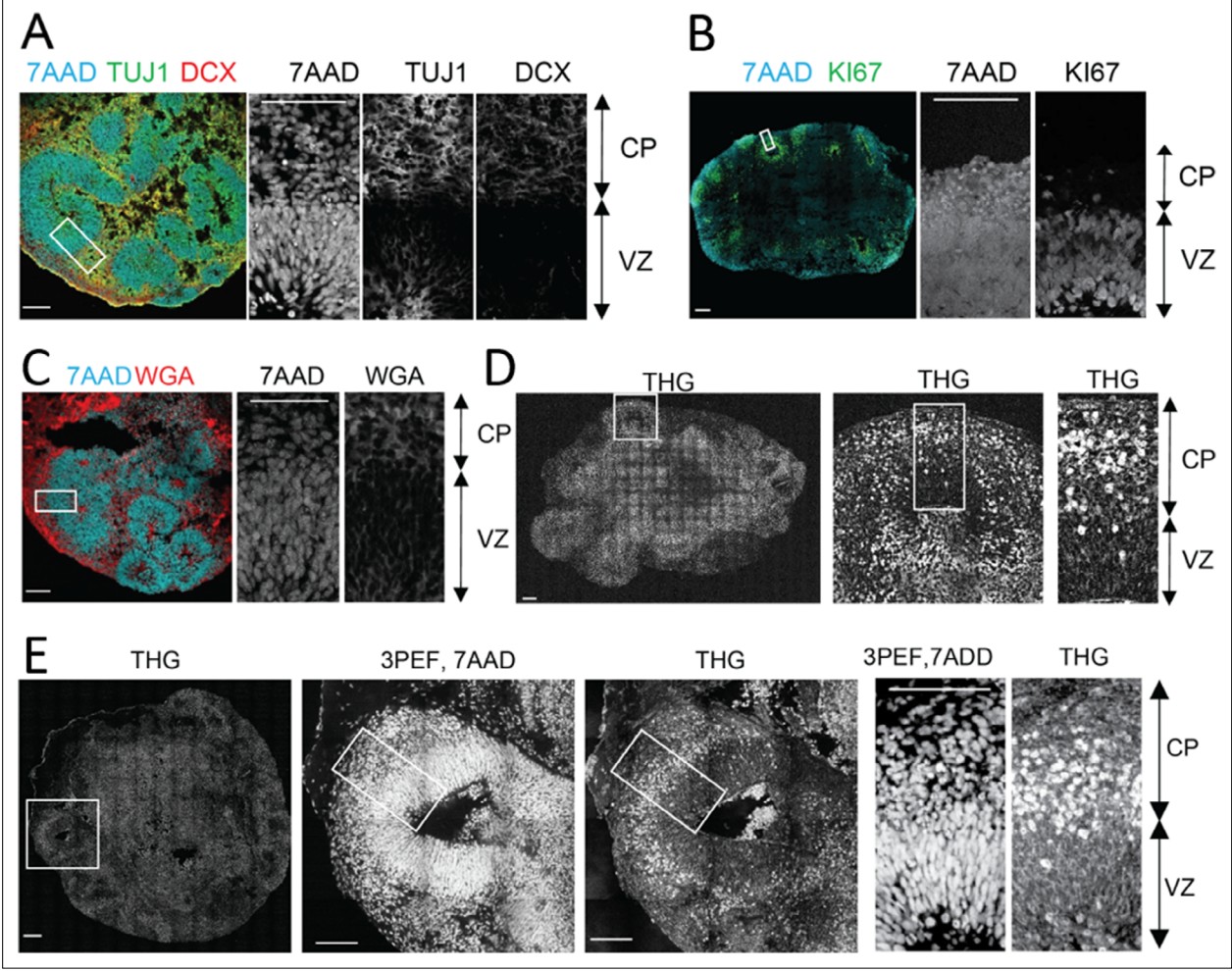

**Figure 2.** Label-free Third Harmonic Generation (THG) imaging of intact cerebral organoids. (**A-B**) Confocal imaging of immunolabeled 2D cerebral organoid slices where cerebral organoids present well-organized spherical structures located around ventricle-like cavities. Around these cavities the ventricular zone-like substructure (VZ), densely populated with progenitors (KI67 + cells) and young migrating neurons (TUJ1 + cells), can be distinguished from the cortical plate-like substructure (CP) with a less dense population of neurons (Line 1, TUJ1, DCX + cells). (**C**) Cortical plate and ventricular zone substructures presented distinct cell density and morphology (Line 2) as shown by nuclear (7AAD dye) and plasma membrane labeling (WGA dye). (**D–E**). Label-free THG imaging from intact fixed 3D cerebral organoids (Line 2) show distinct signals from the cortical plate and ventricular zone, as confirmed by the nuclear labeling (7AAD dye) observed with the same setting but using three-photon epifluorescence (3PEF). THG signal seems to occur at the plasma membranes in the VZ and to be brighter in the CP substructure. Scale bars are 100 µm.

The online version of this article includes the following figure supplement(s) for figure 2:

**Figure supplement 1.** Identifying the identity of THG signals in an intact organoid.

**Figure supplement 2.** Three-dimensional characterization of electroporated cells in a fixed control organoid.

**Figure supplement 3.** The quantitative analysis of GFP, THG, and 7AAD signal overlap in fixed organoids with respect to depth of imaging.

**Figure supplement 4.** Validation of THG signal from identified neurons in organoids.

**Figure supplement 5.** The quantitative analysis of the overlap between GFP and THG signals in live WT and MT organoids.

(*Figure 3B*). The strong THG signal produced at the interface between an organoid'ssurface and the cover glass helped us determine boundaries of the intact organoids. Since our field of view (FOV) was 250 µm, serial imaging of multiple sites was necessary to delineate individual CP regions. Notably, we were able to collect high-resolution THG signal from the entire 900 µm-deep regions, highlighting the performance of three-photon fluorescence and THG detection for deep-tissue imaging, and in particular, intact uncleared whole organoid imaging. The maximum imaging depth was only limited by the working distance of the objective, which was approximately 2 mm in our experiments (*Figure 3— video 3*).

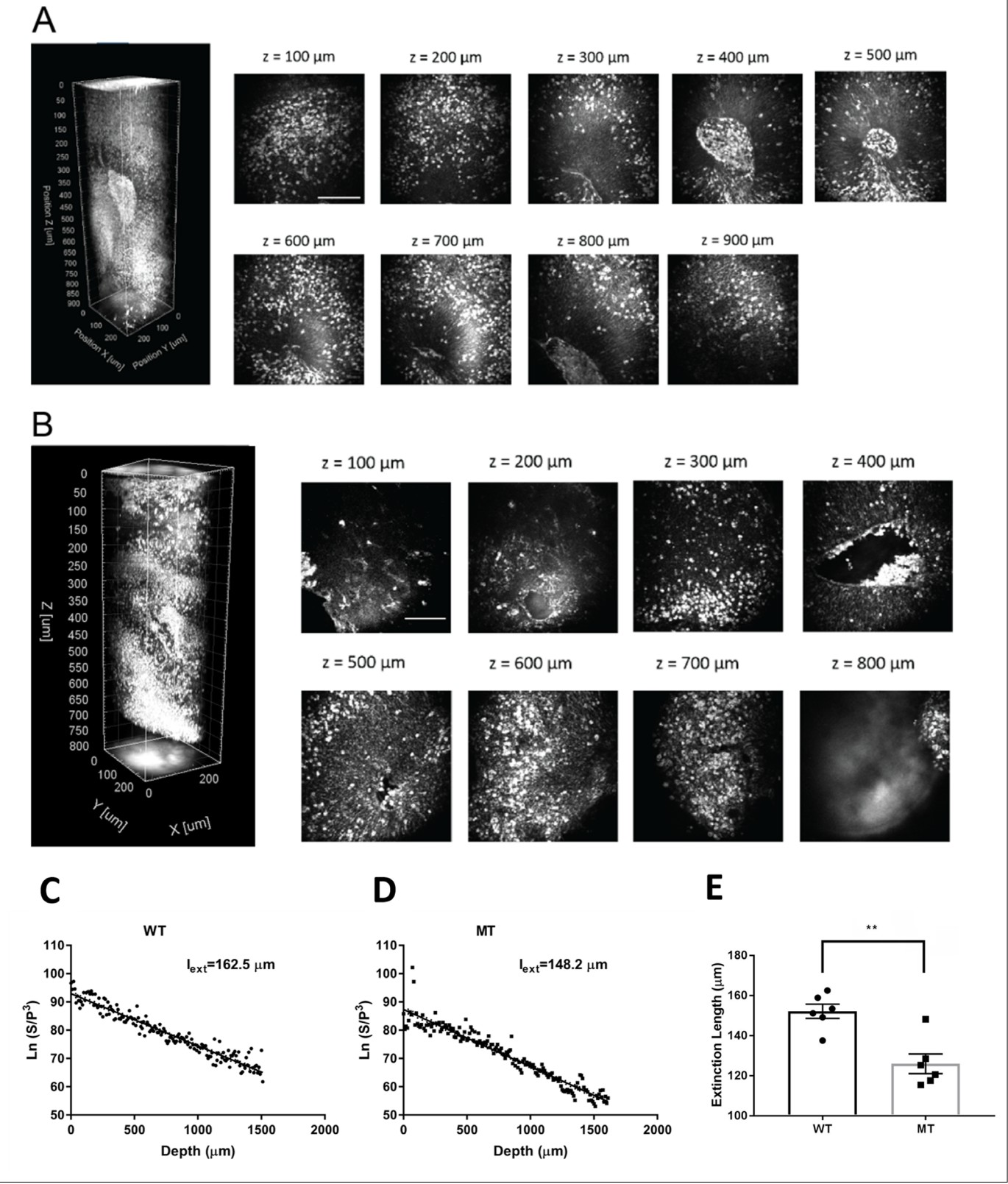

**Figure 3.** 3-D label-free THG imaging of fixed organoids and characterizing their extinction lengths. (**A–B**) 3D reconstruction of label-free THG signal from ventricular regions highlighted in *Figure 2D–E*. THG allows imaging of deep structures (here, till 900 μm). Scale bars represent 100 μm. (**C–D**) Characterization of the extinction lengths of a WT and a MT fixed organoid at 1300 nm excitation wavelength. Semi-logarithmic plot for ratio of PMT signal and cube of laser power with respect to imaging depth for third harmonic generation (THG) imaging. Slope of these curves result in 162.5 μm and

*Figure 3 continued on next page*

*Figure 3 continued*

148.2 µm extinction lengths for the WT and MT fixed organoids, respectively. (**E**) Comparison of the extinction lengths of six WT and MT fixed organoids (Lines 1 and 2) at 1300 nm excitation wavelength. The average extinction length of wild-type (WT) organoids is significantly higher than that of mutant (MT) organoids (n=6 organoids, p<0.05, t-test, error bars are standard error of the mean [SEM]).

The online version of this article includes the following video for figure 3:

**Figure 3—video 1.** Depth-resolved z-stack images of an intact, fixed RTT-MT organoid.

https://elifesciences.org/articles/78079/figures#fig3video1

**Figure 3—video 2.** Depth-resolved z-stack images of an intact, fixed RTT-WT organoid.

https://elifesciences.org/articles/78079/figures#fig3video2

**Figure 3—video 3.** Depth-resolved z-stack THG images up to 2 mm depth of an intact, fixed RTT-WT organoid.

https://elifesciences.org/articles/78079/figures#fig3video3

## Label-free imaging of fixed cerebral organoids reveals differences between RTT and isogenic control organoids

First, we characterized the extinction (combined scattering and absorption) lengths of WT and MT organoids. Our label-free THG imaging characterization shows that the extinction lengths of a fixed WT and a MT organoid are 162.5 µm and 148.2 µm, respectively which are approximately half of the extinction length of a primary visual cortex of an awake mouse brain (*Yildirim et al., 2019*; *Yildirim et al., 2020*; *Figure 3C–D*). In other words, fixed WT and MT organoids are at least two times more scattering than the mouse visual cortex at 1300 nm excitation wavelength. We also examined different WT and MT organoids for comparison (*Figure 3E*). Our results show that average extinction length of WT organoids (152.1±3.5 µm) is significantly higher than that of MT organoids (125.9±4.9 µm, p<0.005, n=6 organoids, t-test).

To study the 3D organization of VZ structures in RTT-WT and RTT-MT organoids, depth-resolved serial imaging of whole organoids was performed automatically with a custom algorithm to reduce the total imaging duration (see Materials and methods). We acquired both 7AAD three-photon fluorescence and THG signal from RTT-WT (*Figure 4A*) and RTT-MT (*Figure 4B*) organoids. Then, we determined the boundaries of individual VZ structures using THG signal and rendered individual cortical regions in both RTT-WT (*Figure 4A*) and RTT-MT (*Figure 4B*) organoids (see Materials and methods, see *Figure 4—video 1*, see *Figure 4—figure supplement 1*). First, we quantified the total volume of each region of interest and found slightly (but not significantly) higher volumes in mutant organoids than in control ones ($1.13\pm0.18 \times 10^7$ µm$^3$ for RTT-WT, and $1.39\pm0.15 \times 10^7$ µm$^3$ for RTT-MT organoids, p=0.3013; n=10 organoids, t-test; *Figure 4C*). However, the surface area of the VZ region in RTT-MT organoids was significantly higher than that in RTT-WT organoids ($5.70\pm0.75 \times 10^5$ µm$^2$ for RTT-WT, and $11.0\pm0.71 \times 10^5$ µm$^2$ for RTT-MT organoids, p<0.0001; n=10 organoids, t-test; *Figure 4C*). In addition, the ratio of volume to area was used as a relative measure of the VZ thickness. The progenitor-rich structures in the RTT-MT organoids had significantly lower unit thickness relative to RTT-WT organoids (23.11±4.12 µm for RTT-WT, and 9.99±0.27 µm for RTT-MT organoids, p<0.0001, n=10 organoids, t-test; *Figure 4C*). Finally, the number of ventricles was significantly higher in RTT-MT organoids (12.3±0.7 for RTT-WT, and 22.0±1.3 for RTT-MT organoids, p<0.0001, n=10 organoids, t-test; *Figure 4C*).

In summary, using label-free intrinsic THG imaging we show that RTT-MT organoids contain an increased number of VZ structures with larger surface area and volume, but lower VZ thickness, compared to RTT-WT organoids (values for each line separately are shown in *Figure 4—figure supplements 2–5*).

## Label-free imaging of live cerebral organoids captures lower migration speed and displacement in RTT organoids

Direct visualization and time-lapse characterization of neuronal migration dynamics with label-free methods have not been examined so far. Thus, we tracked migrating neurons in real time using THG microscopy. We performed live imaging of RTT-WT and RTT-MT organoids at 35 days in vitro (DIV) for 12–96 hr, with a volumetric acquisition interval of 20 min (*Figure 5—figure supplement 1*, see Materials and methods). Given the sparseness and high signal to noise ratio, we were able to track cells harboring a bright somatic THG signal in both RTT-WT and RTT-MT organoids (*Figure 5A*, see

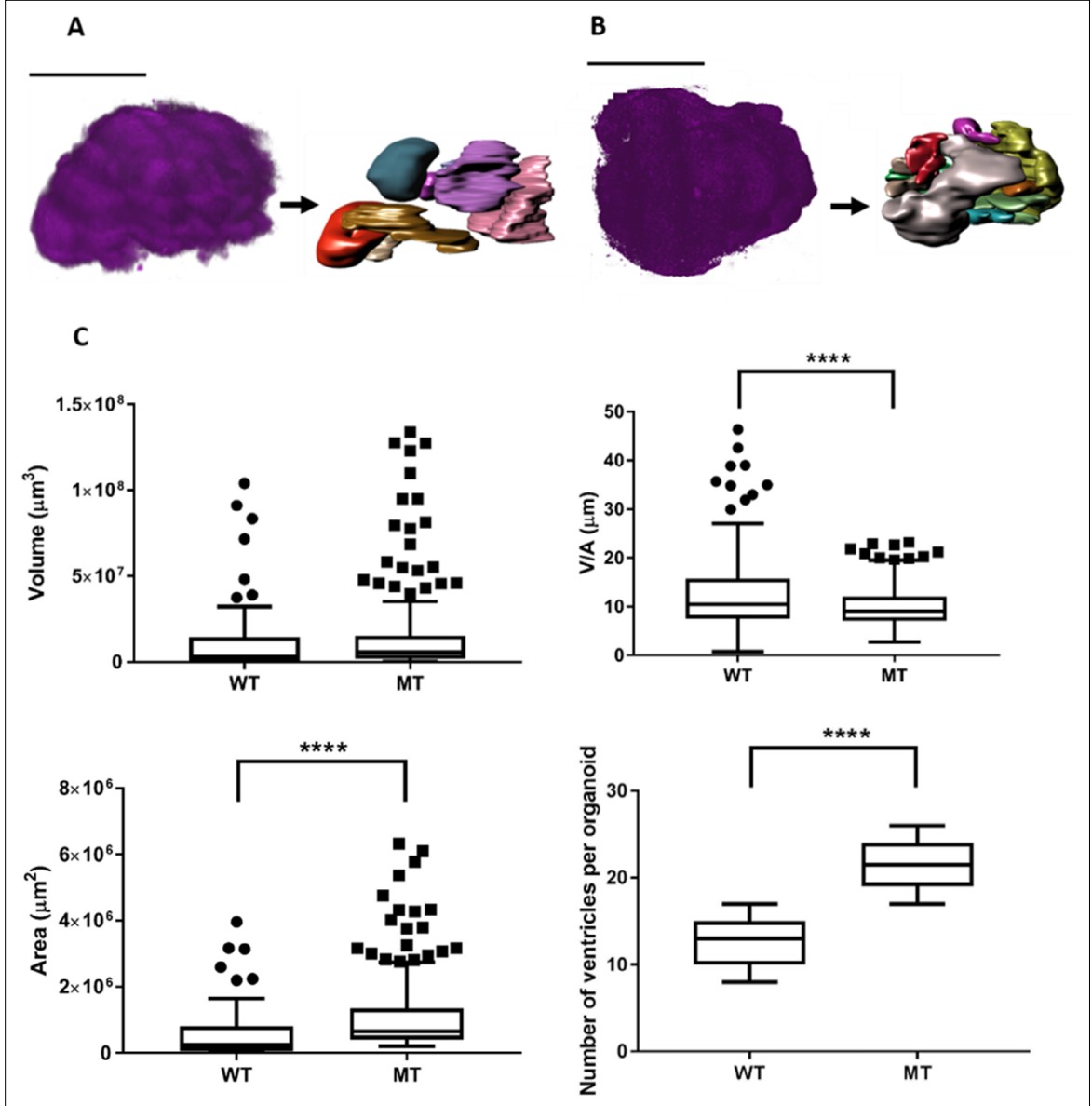

**Figure 4.** Three-dimensional characterization of control (RTT-WT) and mutant (RTT-MT) VZ regions in intact organoids. (**A, B**). THG signal acquired from the whole RTT-WT (**A**) and RTT-MT (**B**) organoids (Line 1) (left) was segmented into individual ventricular regions (right). (**C**) (left; top) RTT-MT organoids showed slightly higher VZ volume than those in RTT-WT organoids ($1.13\pm0.18 \times 10^7$ µm$^3$ for RTT-WT, and $1.39\pm0.15 \times 10^7$ µm$^3$ for RTT-MT organoids, p=0.3013; n=10 organoids, t-test). (left; bottom) The surface area of the VZ region in RTT-MT organoids was significantly higher than that in RTT-WT organoids ($5.70\pm0.75 \times 10^5$ µm$^2$ for RTT-WT, and $11.0\pm0.71 \times 10^5$ µm$^2$ for RTT-MT organoids, p<0.0001; n=10 organoids, t-test). (right; top) The VZ thickness (V/A ratio) in RTT-MT organoids was significantly lower than that in RTT-WT organoids (23.11±4.12 µm for RTT-WT, and 9.99±0.27 µm for RTT-MT organoids, p<0.0001, n=10 organoids, t-test). (right; bottom) The number of ventricles was significantly higher in RTT-MT organoids than that in RTT-WT organoids (12.3±0.7 for RTT-WT, and 22.0±1.3 for RTT-MT organoids, p<0.0001, n=10 organoids, t-test). The data are collected from two lines (Line 1 and Line 2, see *Figure 4—figure supplements 2–5*). Error bars are 90% of the confidence interval (CI).

The online version of this article includes the following video and figure supplement(s) for figure 4:

**Figure supplement 1.** Three-dimensional characterization of control and mutant VZ regions in intact organoids.

**Figure supplement 2.** Comparison of average volume of ventricles per organoid in RTT-WT and RTT-MT organoids for Line 1 and Line 2.

**Figure supplement 3.** Comparison of average surface area of ventricles per organoid in RTT-WT and RTT-MT organoids for Line 1 and Line 2.

**Figure supplement 4.** Comparison of average volume per unit area of ventricles per organoid in RTT-WT and RTT-MT organoids for Line 1 and Line 2.

**Figure supplement 5.** Comparison of average number of ventricles per organoid in RTT-WT and RTT-MT organoids for Line 1 and Line 2.

**Figure 4—video 1.** THG signal acquired form an intact RTT-MT organoid was segmented into individual ventricular regions.
https://elifesciences.org/articles/78079/figures#fig4video1

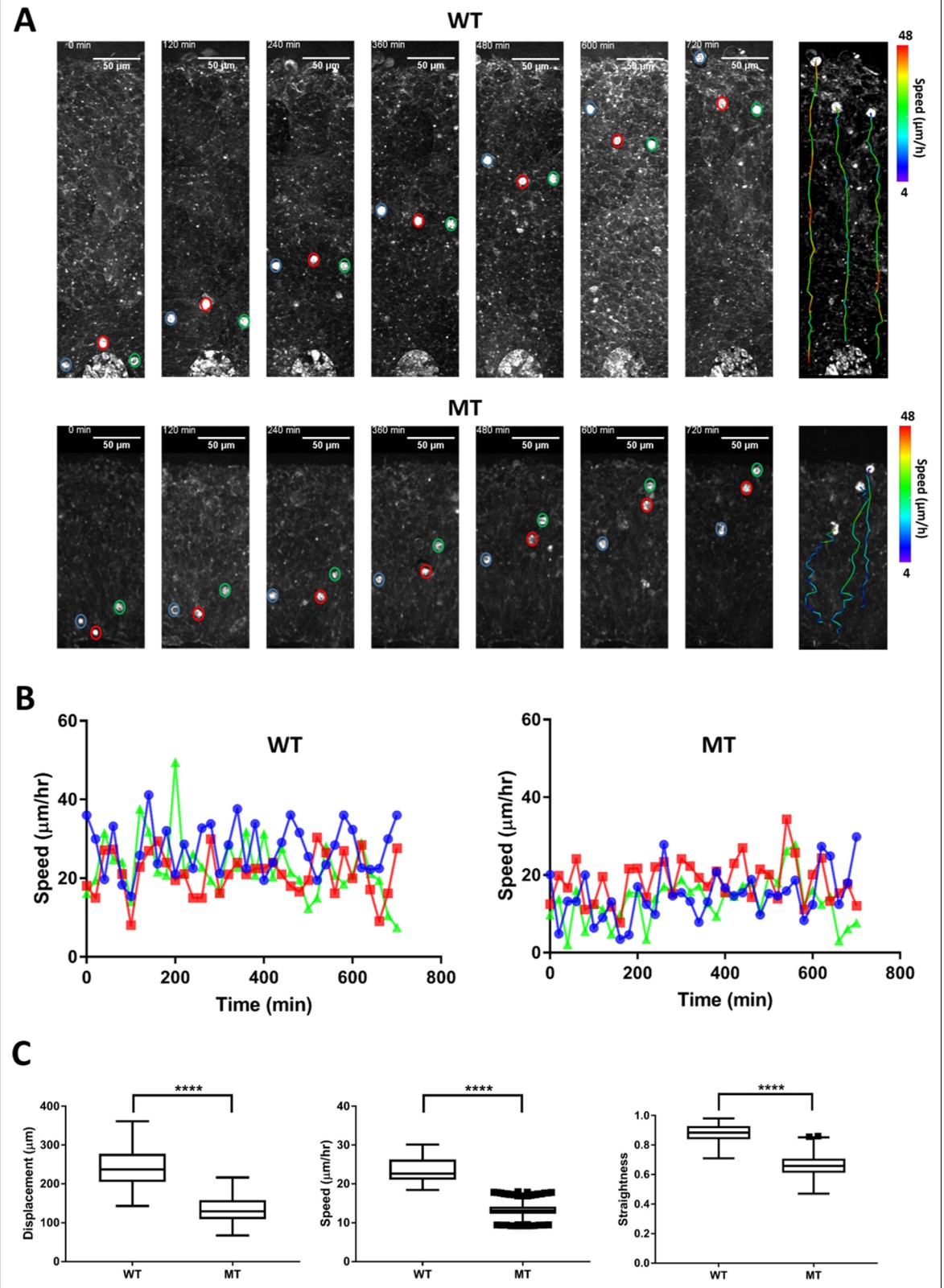

**Figure 5.** Dynamics of neuronal migration in RTT-WT and RTT-MT organoids. (**A**) Representative time-lapse THG images of migrating cells in the ventricular zone in RTT-WT (top) and RTT-MT (bottom) organoids (Line 1) in every 2 hr. The migrating trajectory of each representative cells is shown on the right panel to the time-lapse images. Color bar represents the instantaneous speed of these cells. (**B**) Representative track speed over time for cells in RTT-WT (left) and RTT-MT (right) organoids. (**C**) Summary of displacement, average migration speed and straightness of the migration trajectory (n=6

*Figure 5 continued on next page*

Figure 5 continued

RTT-WT and 6 RTT-MT organoids, 200 cells for WT and 210 cells for MT. Scale bar is 50 µm. **, p<0.01; ***, p<0.001; ****, p<0.0001, t-test). Error bars are 90% of the confidence interval (CI).

The online version of this article includes the following video and figure supplement(s) for figure 5:

**Figure supplement 1.** Representative long-term imaging of live mutant organoid at 1 mm depth.

**Figure supplement 2.** Comparison of average speed of migrating cells in RTT-WT and RTT-MT organoids for Line 1 and Line 2.

**Figure supplement 3.** The distribution of migration speeds of the cells with respect to the depth of imaging in WT and MT organoids.

**Figure supplement 4.** Comparison of the straightness of the migration trajectory of cells in RTT-WT and RTT-MT organoids for Line 1 and Line 2.

**Figure supplement 5.** Comparison of the displacement of migrating cells in RTT-WT and RTT-MT organoids for Line 1 and Line 2.

**Figure supplement 6.** Characterization of cell viability in wild type (WT) and mutant (MT) organoids via Caspase3 immunostaining.

**Figure supplement 7.** Improvement in imaging depth of fixed organoids with refractive index-matching gel.

**Figure 5—video 1.** Neuronal migration in isogenic control (RTT-WT) intact live organoid imaged over the course of 12 hr.
https://elifesciences.org/articles/78079/figures#fig5video1

**Figure 5—video 2.** Neuronal migration in mutant (RTT-MT) intact live organoid imaged over the course of 12 hr.
https://elifesciences.org/articles/78079/figures#fig5video2

*Figure 5—video 1Figure 5—videos 1; 2*, see Materials and methods). Although the tracked cells in both RTT-WT and RTT-MT organoids migrated radially from the inner VZ to the outer VZ, RTT-MT cells showed decreased displacement, more tortuous trajectories, and reduced linear speed than RTT-WT cells (*Figure 5B and C*, also see *Figure 6*). Specifically, we observed in RTT-WT organoids an average migration speed of 23.4±0.2 µm/hr (n=6 organoids, N=208 cells). The average speed for each line was 20.5±0.1 µm/hr for Line 1 and 25.6±0.2 µm/hr for Line 2 (*Figure 5—figure supplement 2*). These average speed values are comparable to the speed recorded in neurons migrating off a human wild-type cerebral organoid onto a Matrigel-coated surface (*Bershteyn et al., 2017*) and in ferret cortical explants (*Gertz and Kriegstein, 2015*). In contrast, RTT-MT cells had a reduced average speed of 13.3±0.1 µm/hour (n=6 organoids, N=217 cells). The average speed for each line was 11.4±0.1 µm/hour for Line 1 and 14.5±0.1 µm/hr for Line 2 (*Figure 5—figure supplement 2*). Collectively, the average speed in WT organoids was significantly higher than the average speed in MT organoids (*Figure 5C*). In addition, we compared migration speeds of individual cells with respect to their location in each organoid as well as between organoids (*Figure 5—figure supplement 3*). In conclusion, we did not find any statistically significant difference in migration speeds of individual cells at different locations in the same organoid (*Figure 5—figure supplement 3A-B*) as well as between organoids of the same type (*Figure 5—figure supplement 3C-D*). We also quantified the straightness of trajectory by taking the ratio of displacement and total trajectory length. With a straightness value close to 1, RTT-WT cells (straightness value of 0.881±0.004) had significantly more straight trajectories than RTT-MT cells (0.658±0.006) which showed much more nonlinear migrating patterns (*Figure 5A and C*).

Since the average speed of RTT-MT was lower and these cells had a less straight trajectory, net displacement of MT cells (133.2±2.4 µm) was significantly lower than that of WT cells (244.4±3.6 µm) (*Figure 5C*). The straightness and net displacement values of RTT-WT and RTT-MT in both lines showed similar trends (*Figure 5—figure supplements 4 and 5*). To check whether cells in the organoids were still viable after imaging session of ~18 hr, we performed Caspase3 immunostaining for both imaged and not-imaged control organoids as well as a 90-day-old organoid with a necrotic core as a positive control (*Figure 5—figure supplement 6*). Our results show that imaging live organoids in the mini-incubator with three-photon microscopy did not induce any significant cell death compared to that in organoids that were not kept in the mini-incubator or imaged (*Figure 5—figure supplement 6*).

Electroporation of organoids with GFP construct is a powerful method to label migrating neurons with high efficiency and low toxicity (*Mellios et al., 2011*). We performed a comparative live-cell imaging in GFP-electroporated wild-type and mutant organoids and confirmed that most cells expressing GFP also emitted prominent THG signal (*Figure 2—figure supplement 4A-B*). Indeed, we found that 23/25 GFP-positive cells in the WT organoids and 27/30 cells in the MT organoids emitted somatic THG signals (*Figure 2—figure supplement 5*). Cells expressing both GFP and THG

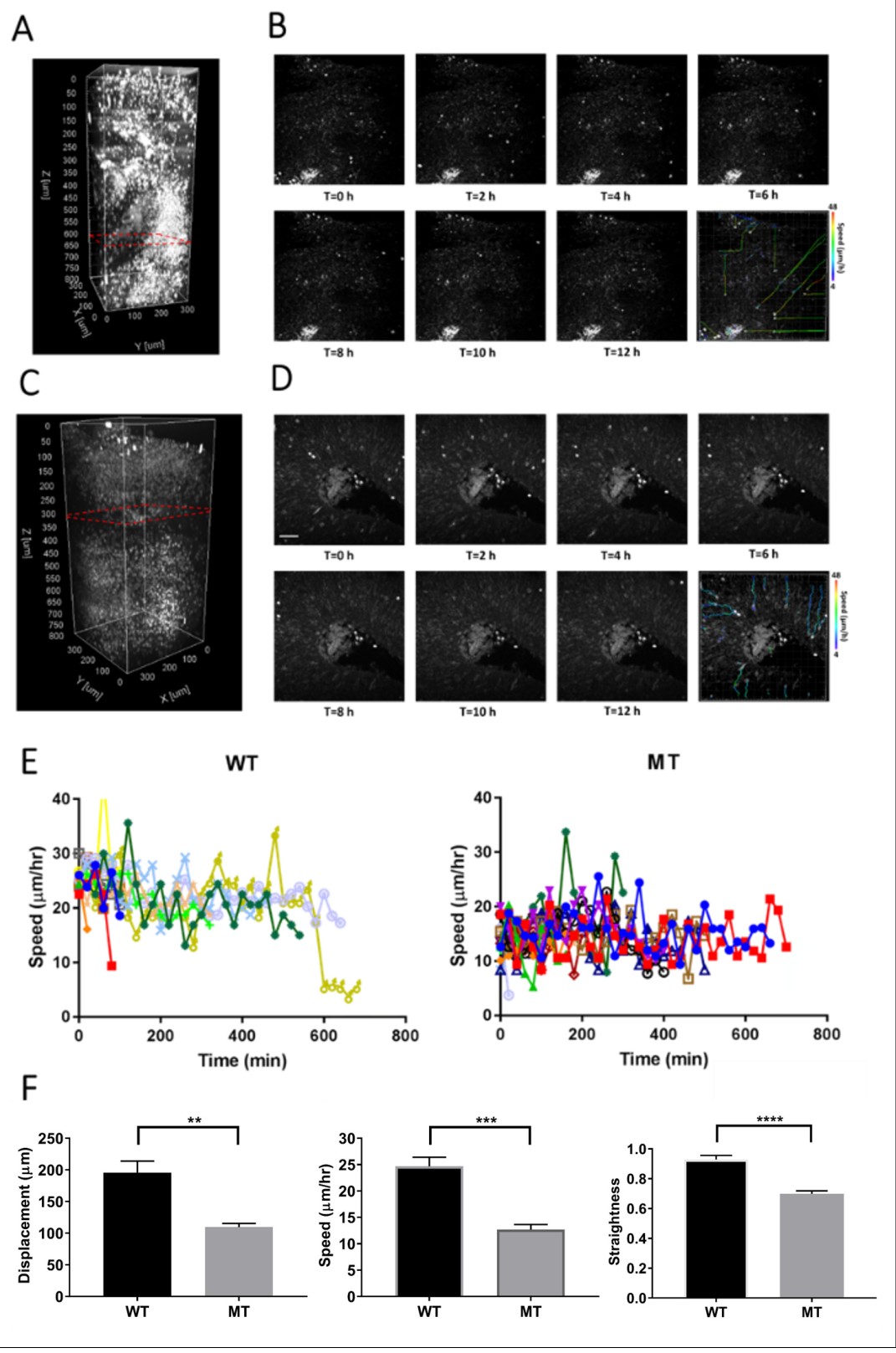

**Figure 6.** Dynamics of neuronal migration in RTT-WT and RTT-MT organoids. Representative three-dimensional THG images of the live RTT-WT (**A**) and RTT-MT (**C**) organoids from Line 2. Dashed red lines show the planes of imaging presented in (**B**) and (**D**). Representative time-lapse, depth-resolved THG images of migrating cells in the VZ region for RTT-WT (**B**) and RTT-MT (**D**) organoids in every 2 hr. The migrating trajectory of each representative

*Figure 6 continued on next page*

*Figure 6 continued*

cells is shown on the bottom right panel to the time-lapse images. Color bar represents the instantaneous speed of these cells. (**E**) Representative track speed over time for cells in RTT-WT and RTT-MT organoids. (**F**) Summary of displacement, average migration speed and straightness of the migration trajectory. Scale bar is 50 μm.

signal demonstrated similar migration deficits in their migration speeds (*Figure 2—figure supplement 4C-D*) as described above (*Figures 5–6*).

## Discussion

Since the demonstration that neural differentiation of human induced pluripotent stem cells could be induced in 3D and used to recapitulate the early stages of embryonic brain development, cerebral organoids have raised considerable interest as a model for understanding embryonic stages of cortical development and neurodevelopmental disorders (*Lancaster et al., 2013*; *Qian et al., 2016*; *Paşca et al., 2015*; *Mariani et al., 2012*; *Kadoshima et al., 2013*; *Luo et al., 2016*; *Camp et al., 2015*; *Zhang et al., 2019*; *Bershteyn et al., 2017*; *Klaus et al., 2019*; *Garcez et al., 2016*; *Dang et al., 2016*; *Mariani et al., 2015*; *Ip et al., 2018*) However, high-resolution 3D imaging of intact and uncleared organoids is challenging due to high light scattering. Thus, the majority of analyses of organoids has been limited to imaging tissue sections or intact organoids, with one-photon microscopy (*Lancaster et al., 2013*; *Paşca et al., 2015*; *Bershteyn et al., 2017*; *Klaus et al., 2019*; *Birey et al., 2017*; *Birey et al., 2022*; *Xiang et al., 2017*) or imaging superficial parts of intact organoids with two-photon microscopy (*Mansour et al., 2018*). Especially for live cell imaging, the one photon imaging depth is limited to 20 μm, with improvement to 40 μm with tunable refractive-index matching medium (*Boothe et al., 2017*). Two photon imaging depth is limited to a couple of hundred micrometers with live cerebral organoids (*Mansour et al., 2018*). To image intact and uncleared organoids, we thus developed custom three-photon microscopy to perform both fluorescence and label free (THG) three-photon imaging. Here, we demonstrate for the first time that three-photon fluorescence and THG microscopy is capable of high spatial resolution imaging deep within intact human cerebral organoids, enabling 3D imaging of intact cerebral organoids as well as long-term live cell time-lapse imaging of radially migrating neurons in intact cerebral organoids. Since we used moderate repetition rate of the laser (800 kHz), uniquely reduced the pulse width <30 fs in the deepest parts of the organoid for the first time (see Materials and methods), and optimized the intermediate optics in the excitation and emission paths, the average power on the surface of the organoids did not exceed 5 mW, which induced no noticeable tissue damage such as cell death (*Figure 5—figure supplement 6*). No specific processing of the fixed organoids, such as clearing techniques (*Murray et al., 2015*; *Chung and Deisseroth, 2013*) was necessary; however, incubation in index matching solution improved the imaging depth if necessary (*Figure 5—figure supplement 7*).

Most of the THG signal we observed in RTT-WT and RTT-MT organoids arose from cellular membranes or showed a high-intensity somatic profile observed mostly in the CP region. It was shown previously that the cell nucleus (*Lin et al., 2014*; *Tsai et al., 2011*) and several organelles in the cell cytoplasm could generate a THG signal (*Yelin and Silberberg, 1999*; *Barzda et al., 2005*). Thus, we propose that the bright somatic signal is the result of the high concentration of organelles in the neuronal soma. THG imaging of intact fixed organoids provided clear demarcation between VZ and CP structures, likely due to differences in cell composition, morphology and density in these regions. This clear label-free demarcation of VZ and CP further allowed us to characterize VZ 3D structures in RTT-WT and RTT-MT organoids. We characterized the extinction lengths of RTT-WT and RTT-MT organoids (*Figure 3C–E*) and found that average extinction length of WT organoids is significantly higher than that of MT organoids. We suggest that higher surface area with lower radial thickness in MT organoids resulted in irregular shapes of ventricular zones. In addition, the number of ventricular zones in MT organoids is significantly higher than in WT organoids. Therefore, higher number of irregular shaped ventricular zones may induce more scattering in MT organoids which results in their lower extinction lengths. Moreover, a bright THG signal arose from neuronal soma, allowing live cell tracking of migrating neurons in the VZ. To check the source of THG signal in the VZ, we labeled migrating neurons with GFP through electroporation. We showed that more than 85% of the GFP positive cells had also significant THG signal, suggesting that label-free THG imaging of live cerebral organoids can

reveal the migration trajectories and deficits at the single-cell level (*Figure 2—figure supplements 2–5*). In addition, we also performed DCX staining to the same field of view that we imaged with our system. This staining also shows that THG signal is mostly overlapped with DCX signal which indicates that THG signal is mostly populated in the cortical plate but not in the ventricular zone (*Figure 2—figure supplement 1*).

Our findings of 3D morphological alterations in RTT-MT organoids including an increase in total ventricular zone volume, a significant increase in ventricular zone surface area, and a significant decrease in mean ventricle thickness, enable noninvasive and comprehensive quantification of organoids consistent with enhanced proliferation of the progenitor pool (*Mellios et al., 2011*). As VZ expansion is uniquely different in human versus mouse neurogenesis (*Florio and Huttner, 2014*), it is likely that this phenotype would go unobserved in mouse (or other lissencephalic) models of RTT. Moreover, we and others previously suggested with endpoint experiments that the putative migration of young neurons through the VZ is impaired in mice after in utero electroporation of Mecp2-sh RNA as well as in human RTT cerebral organoids (*Mellios et al., 2011*; *Bedogni et al., 2016*). However, the dynamics of the migration deficit were never previously characterized. Thus, here we performed label-free live cell tracking in intact and uncleared human cerebral organoids at different depths and locations to explore the dynamics of neuronal radial migration in RTT-WT and RTT-MT organoids. RTT-MT neurons demonstrated reduced migration speed and less straight trajectories and consequently a lowered displacement than RTT-WT cells in cerebral organoids at 5 weeks. Furthermore, these trends in migration speed were independent of the imaging depth or the imaging location in both MT and WT organoids. Although there was a range of radial thickness in MT organoids, lower migration speed and tortuous migration patterns were common. Altered cytoskeleton dynamics have been previously suggested in rat, mouse and human models of Rett syndrome and could contribute to migration defects (*Kaufmann et al., 1995*; *Delépine et al., 2013*; *Delépine et al., 2016*; *Cortelazzo et al., 2014*; *Bhattacherjee et al., 2017*). Non-cell autonomous effects could also contribute to the neuronal radial migration deficit, such as impaired cell-to-cell communication, adhesion and radial glia dysfunction. Further studies are required to study the mechanisms responsible for the migration deficit as well as to rescue the phenotype. Taken together, these results suggest that changes in ventricular wall thickness result not only from the decreased neurogenesis and increased proliferation of the progenitor pool but also from the abnormally slow displacement and tortuous trajectory of the neurons. Importantly, our findings directly demonstrate dynamic deficits in migration at very early stages of prenatal development which provide insight into previously described deficits in human RTT brains, including reduced cortical thickness, cell size and dendritic arborization, and reduced cerebral volume (*Armstrong et al., 1995*; *Bauman et al., 1995*; *Carter et al., 2008*).

Our study demonstrates the promise of three-photon and label-free THG imaging for monitoring spatiotemporal dynamics in intact and uncleared human cerebral organoids. We believe that our optimized three-photon microscopy system is straightforward to use, and can provide a new platform for monitoring structural and functional abnormalities in fixed and live organoids derived from patients with neurogenetic disorders as well as for therapeutic drug screening. In addition, we can study phenotypes in calcium activity in neurons especially VZ calcium dynamics such as waves (*Weissman et al., 2004*) as well as their migration deficits in cerebral organoids. In addition, we can capture structural phenotypes in myelinated axons and endothelial cells (microvasculature; *Yildirim et al., 2020*). We believe that imaging myelinated axons as well as endothelial cells can be used for studying brain disorders such as Multiple Sclerosis and Alzheimer's disease with cerebral organoids since there is a significant loss of myelin as well as reduction of blood flow in these diseases, respectively. Finally, it should be possible to perform structural and functional three-photon and THG imaging for revealing structural and functional phenotypes for cancer applications such as glioblastoma (*Bakker et al., 2022*).

## Materials and methods
### Custom microscope for three-photon and THG imaging
To obtain an excitation wavelength of 1300 nm required for label-free THG imaging deep within intact organoids, ultrashort laser pulses (300 fs, 400 kHz,16W) at 1045 nm from a pump laser (Spirit, Spectra Physics) were sent to a noncollinear optical parametric amplifier (NOPA, Spectra Physics). In order to

shorten the pulse width on the sample to maximize the generated THG signal, we built a two-prism based external compressor to prechirp the pulse before sending it to the microscope. The internal compressor in NOPA could compress the Gaussian pulse width to 22 fs (which was the transform limited pulse width for 110 nm spectral full width half maximum bandwidth). Due to the dispersion in the microscope, the Gaussian pulse width on the sample was ~250 fs. After dispersing the pulse with 65 cm long prism-pair, and silicon broadband window (Thorlabs, WG81050) the pulse width on the sample was reduced to 27 fs in the deepest part of the organoids (pulse widths <30 fs obtained, to our knowledge, for the first time). To measure the pulse width inside the organoids, we placed a live organoid (<1 mm thickness) on the external sensor of the autocorrelator (CARPE). Then, we measured the pulse width of the laser after it exits from the thick organoid. In addition, we built a delay line to double the repetition rate thereby increasing the frame rate of THG imaging. Subsequently, the frame rate at 512 × 512 pixel resolution was increased up to 2 Hz. Power control was performed with a combination of half-wave plate (AHWP05M-1600, Thorlabs) and low-GDD ultrafast beamsplitter (UFBS2080, Thorlabs) with 100:1 extinction ratio. The laser beams were scanned by a pair of galvo-nometric mirrors (6215 H, Cambridge Technologies) to image the laser spot on the back aperture of the objective using a pair of custom-designed scan and tube lenses. A pair of collection lenses for three photon-multiplying tubes (PMT) collected the emitted signal from the cerebral organoids. To maximize the signal detected in PMTs, optics in both excitation and emission paths were optimized considering the objective lens using Zemax software. Fluorescent signals from immunostaining labels (7AAD) were detected using GaAsP photomultiplier tubes (H7422A-40, Hamamatsu, Japan) and a bandpass filter (Semrock, 650/70 nm), and THG signal was detected using bialkali (BA) photomultiplier tube (R7600U-200) and a narrow-bandwidth bandpass filter (430/20 nm). A custom designed objective lens (25 x, 1.05 N.A., Olympus) was used. Automated image acquisition, control of scanners, and sample stage were carried out using ScanImage (Vidrio) (*Pologruto et al., 2003*). Imaged cells were located at a depth of 0–2000 μm below the surface of cerebral organoids. The images were taken with 2 μm and 5 μm axial increments in fixed and live organoids, respectively. In live cell imaging, we acquired 50–100 z-stack images to make a volumetric acquisition. This volumetric imaging took approximately 3–6 min and we repeated it every 20 min. Laser power ranged from 0.5 to 5 mW at the sample depending on depth and fluorescence expression levels. Both fixed and live organoids were placed on a two-axis motorized stage (MMBP, Scientifica) and the objective lens was placed on a single-axis motorized stage (MMBP, Scientifica) to move it in the axial direction.

## Methods for fixed and live organoid imaging

For fixed organoid imaging, five-week-old cerebral organoids were placed into an imaging chamber gasket (C18161, ThermoFisher) filled with index-matching medium Hprotos (*Murray et al., 2015*). Top and bottom parts of the imaging chamber gasket were sandwiched between a glass slide (48311–703, VWR) and a cover glass (48366–227, VWR) (*Figure 1B*). To perform whole fixed organoid imaging, automatic serial imaging of multiple sites was performed via determining the coordinates of four corners of the entire field of view. A Matlab-based algorithm moved the stage in the required x and y axis locations to image the full field of view and then moved the focal plane in the z direction.

For live organoid imaging, we increased the imaging field of view from 250 μm to 320 μm. The 5-week-old live organoids were placed in a microincubator (RC-30, Warner Instruments). The microincubator temperature was controlled and maintained at 37 °C with a pair of resistive heater elements. A peristaltic pump was connected to the microincubator to provide continuous flow of cell medium supplemented with 5% $CO_2$, 5%$O_2$, and balanced $N_2$ (Airgas) (*Figure 1C*). We performed Caspase3 immunostaining to check whether our imaging system induced any significant cell death. Compared with healthy and not-imaged organoids, the organoids which were imaged up to 18 hr in our imaging system did not show any increased cell death (*Figure 5—figure supplement 6*).

## iPSC generation and colony maintenance

Induced pluripotent stem cells (iPSC) were reprogrammed from Rett Syndrome patient samples obtained from the Coriell cell line repository (*Mellios et al., 2011*; *Nott et al., 2016*). These lines were derived from: (1) a patient carrying a MECP2 V247X mutation (Coriell Institute biobank GM07982); (2) a patient with MECP2 R306C mutation (Coriell Institute biobank GM23298). All iPSC lines were primarily and routinely quality-controlled by karyotyping (G-band), mycoplasma testing

and pluripotency testing (initially by teratoma test then routinely by immunostaining for pluripotency markers such as Sox2 and Oct4). We examined allele-specific transcription of the MECP2 gene in all the iPSC lines by performing RNA extraction, reverse transcription, PCR and Sanger sequencing of the amplified fragments. In all lines we were able to detect transcription of either the wildtype or the mutant alleles of the MECP2 gene, but not both, confirming the clonality and X inactivation status. We confirmed that the X inactivation profile is maintained at all stages of the organoid differentiation. Cell culture quality check and authentication were performed every ten iPSC passages. iPSC colonies were grown in iPSC media, consisting of 20% Knock-out Serum Replacement (KOSR) (Invitrogen), 1% penicillin/streptomycin (Invitrogen), 1% non-essential amino acids (Invitrogen), 0.5% L-glutamine (Invitrogen), 100 µM 2-mercaptoethanol (Bio-Rad), DMEM/F-12 (Invitrogen), supplemented with 10 ng/mL bFGF (Stemgent). Culture media was changed daily. iPSC colonies were passaged weekly onto 6-well plates coated with 0.1% gelatin (EMD Millipore) and pre-seeded with a feeder layer of irradiated mouse embryonic fibroblasts (MEFs) (GlobalStem), which were plated at a density of 200,000 cells/well. Passaging entailed lifting colonies with 2.5 mg/ml Collagenase, Type IV (ThermoFisher) for 1 hr and dissociation into smaller pieces through manual tritulation before seeding onto feeder layer of MEFs.

## Cerebral organoid generation

iPSCs were detached from irradiated MEFs and plated at $9 \times 10^4$ cells per well of an ultralow attachment 96-well plate (Corning) in iPSC media supplemented with bFGF (10 ng/mL) and ROCK inhibitor (50 µM; Y-27632, Tocris) (Day 0). Embryoid bodies (EBs) were subsequently transferred (Day 6) in an ultra-low attachment 24-well plate (Corning) to neural induction media: 1% N2 supplement (Invitrogen), 1% Glutamax (Invitrogen), 1% non-essential amino acids (Invitrogen), 5 µg/mL heparin (Sigma), DMEM/F12 (Invitrogen), supplemented with 10 µM SB431542 (Tocris Bioscience) and 1 µM dorsomorphin (Stemgent). EBs were embedded in Matrigel (Corning) droplets on Day 11 and transferred to neural differentiation media: DMEM/F12: Neurobasal (Invitrogen), 0.5% N2 supplement, 1% Glutamax, 0.5% non-essential amino acids, 100 µM 2-mercaptoethanol, insulin, 1% Pen/Strep (Invitrogen) supplemented with 1% B27 without vitamin A (Gibco, Life Technologies). On Day 15, embedded EBs were transferred to a shaker and grown in neural differentiation media supplemented with B27 with vitamin A (Gibco, Life Technologies). Organoids were either fixed on Day 35 by a 3-hr incubation in 4% paraformaldehyde solution or imaged under the microscope for live cell characterization.

## Cryosectioning and immunostaining

For cryosectioning, fixed organoids were incubated in sucrose solution overnight at 4 °C before embedding and freezing in O.C.T medium. Frozen organoid tissue was sliced into 20 µm sections using a cryostat. Permeabilization/blocking was performed using 3% BSA/0.1% TX100 in TBS. Incubation of sections from cerebral organoids in primary antibodies solution was performed overnight at 4 °C and in secondary antibodies solution at room temperature for 1 hr (Alexa Fluor, Molecular Probes). The following primary antibodies were used: DCX (Aves Labs, 1:200); PAX6 (Millipore; 1:200); TBR1 (Abcam 1:400); Cleaved capsase-3 (Cell Signaling, 1:500). Coverslips were affixed with ProLong Gold antifade reagent with DAPI (Life Technologies) and z-stacks were acquired using either a Leica TCS SP8 confocal microscope.

## Intact organoid staining

For 3D imaging of whole organoids, intact fixed organoids were incubated with a 1:50 solution of the nuclear dye 7-Aminoactinomycin D (7AAD, MolecularProbes) in PBS for 1 hr at room temperature. Before imaging, organoids were transferred to a shaker and immersed twice for 3 hr in index matching solution (*Murray et al., 2015*). Z-stacks were acquired with the three-photon microscope described above.

## Electroporation

Organoids (specifically, the 'ventricles' or progenitor zones) were electroporated with a GFP construct (pCAG-GFP) at 7 weeks post-EB formation. Plasmids were maxi-prepared using Qiagen Maxi kit. DNA solution (0.5 µg/µl) was mixed with Fast Green dye (0.1%) and injected into the organoids using a pulled-glass capillary microelectrode. Successful injection was confirmed by the visualization of Fast

Green dye inside the injected organoid. Immediately after DNA injection, four 50 ms electrical pulses (40 V) were applied at 1 s intervals using a 5 mm electrode and an electroporator (EM830, BTX). The organoids were returned to the incubator after electroporation. After 3–4 days, the electroporated organoids were subjected to the three-photon microscope for imaging.

### Statistical analysis

All the statistical comparisons presented in the figures and figure supplements (*Figure 3E*, *Figure 4C*, *Figure 5C*, *Figure 6F*, *Figure 4—figure supplements 2–5*, and *Figure 5—figure supplement 2*; *Figure 5—figure supplement 4*; *Figure 5—figure supplement 5*) are based on t-test method where the error bars are 90% of the confidence interval (CI). Points below and above this confidence interval are drawn as individual dots.

### Area segmentation and rendering

The area segmentation was performed manually for individual 2-D THG images and confirmed with 7AAD images which are taken as ground truth (*Figure 4A–B*). Then, Surface function of Imaris software was used to render these 2-D segmented areas into a 3-D volume. We used both opaque (*Figure 4A–B*) and transparent modes (*Figure 4—figure supplement 1C-D*) of this function in order to represent 3-D structure of ventricular zones.

### Live cell tracking

The three-dimensional THG images were used to characterize the speed, straightness, and displacement of cells in control and mutant organoids. To quantify these features of control and mutant cells, Spots function of Imaris software was used. This function first identifies the location of the individual cell, and tracks them in time-lapse images.

### Colocalization of GFP, THG, and 7AAD signals

The colocalization of GFP, THG, and 7AAD signals were performed through Coloc module of Imaris software. We used our z-stack 2-D images (~1,000 images) to form three-channel three-dimensional data sets (*Figure 2—figure supplement 2*). Then, Coloc module quantified the percentage of overlapping between each channel.

### Incubation of fixed organoids in index-matching solution

For imaging some of the fixed organoids, we incubated organoids in the H-protos index- matching solution for 3 hr. With this incubation, we observed improvement in maximum imaging depth up to 50% compared to the condition without the incubation (*Figure 5—figure supplement 7*).

## Acknowledgements

This work was supported by US National Institute of Health (NIH) grants MH085802 and NS090473 (MS), 4-P41-EB015871 (PTCS), K99EB027706 (MY), US National Science Foundation (NSF) grant EF1451125 (MS), a Picower Institute Engineering Collaboration Grant (MS, PTCS and MY), and an equipment grant from the Massachusetts Life Sciences Initiative. AN was supported by the UK Dementia Research Institute, which receives its funding from UK DRI Ltd, funded by the UK Medical Research Council, Alzheimer's Society, and Alzheimer's Research UK. LHT was supported by JPB Foundation. G-lM was supported by National Institutes of Health grant R35NS097370. We thank Dr. Alex Albenese and Dr. Kwunghun Chung for their assistance on Hprotos generation. We are grateful to members of the Sur and So labs for their help.

## Additional information

### Funding

| Funder | Grant reference number | Author |
|---|---|---|
| National Institute of Biomedical Imaging and Bioengineering | K99EB027706 | Murat Yildirim |
| National Institute of Mental Health | MH085802 | Mriganka Sur |
| National Institute of Mental Health | NS090473 | Mriganka Sur |
| National Institute of Biomedical Imaging and Bioengineering | P41-EB015871 | Peter TC So |
| National Science Foundation | EF1451125 | Mriganka Sur |
| National Institute of Neurological Disorders and Stroke | R35NS097370 | Guo-Li Ming |
| JPB Foundation | | Li-Huei Tsai |

The funders had no role in study design, data collection and interpretation, or the decision to submit the work for publication.

### Author contributions

Murat Yildirim, Conceptualization, Software, Formal analysis, Funding acquisition, Methodology, Writing – original draft; Chloe Delepine, Methodology, Writing – original draft, Generated cerebral organoids; Danielle Feldman, Methodology, Generated cerebral organoids; Vincent A Pham, Methodology, Generated cerebral organoids; Stephanie Chou, Methodology, Generated cerebral organoids; Jacque Ip, Methodology, Carried out organoid electroporation; Alexi Nott, Methodology, Provided cell lines; Li-Huei Tsai, Methodology, Provided cell lines; Guo-Li Ming, Methodology, Provided cell lines; Peter TC So, Methodology, Writing – original draft, Writing - review and editing; Mriganka Sur, Resources, Supervision, Funding acquisition, Writing – original draft, Writing - review and editing

### Author ORCIDs

Murat Yildirim (iD) http://orcid.org/0000-0003-0853-2557
Chloe Delepine (iD) http://orcid.org/0000-0002-1856-9583
Vincent A Pham (iD) http://orcid.org/0000-0001-7083-9360
Alexi Nott (iD) http://orcid.org/0000-0002-2029-7193
Li-Huei Tsai (iD) http://orcid.org/0000-0003-1262-0592
Mriganka Sur (iD) http://orcid.org/0000-0003-2442-5671

### Decision letter and Author response

Decision letter https://doi.org/10.7554/eLife.78079.sa1
Author response https://doi.org/10.7554/eLife.78079.sa2

## Additional files

### Supplementary files
• Transparent reporting form

### Data availability

All raw data has been deposited to Dryad (https://dx.doi.org/10.5061/dryad.hqbzkh1jz) and is freely available.

The following dataset was generated:

| Author(s) | Year | Dataset title | Dataset URL | Database and Identifier |
|---|---|---|---|---|
| Yildirim M, Delepine C, Feldman D, Pham V, Chou S, Ip J, Nott A, Tsai L, Ming G, So P, Sur M | 2022 | Label-free three-photon imaging of intact human cerebral organoids: Tracking early events in brain development and deficits in Rett Syndrome | https://doi.org/10.5061/dryad.hqbzkh1jz | Dryad Digital Repository, 10.5061/dryad.hqbzkh1jz |

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
