## [Editor Report]

This manuscript will be of interest to stem cell and developmental biologists who aim to use newly emerging brain organoid models to understand the structure and function of the developing human brain. It presents a technological advance in imaging and describes an innovative method for labeling and tracking cells within organoids to enable the assessment of dynamic processes within the intact organoid. The method is validated in a disease model and addresses a challenge in the field of human stem cell modeling of assessing cells within the 3D structure.

---

## [Decision Letter]

**Decision letter after peer review:**

Thank you for submitting your article "Label-free three-photon imaging of intact human cerebral organoids: tracking early events in brain development and deficits in Rett Syndrome" for consideration by *eLife*. Your article has been reviewed by 3 peer reviewers, and the evaluation has been overseen by a Reviewing Editor and Jeannie Chin as the Senior Editor. The following individuals involved in the review of your submission have agreed to reveal their identity: Anita Bhattacharyya (Reviewer #1); Simon R Schultz (Reviewer #2).

Essential revisions:

1) The prior literature in this area of research is not put in perspective well to reflect the current approaches in organoid imaging. There is a significant body of literature describing migration imaging using brain organoid models, including ones that use intact (not sliced) assembloid preparations not cited in the paper, as opposed to what's been suggested by the authors (Lines 60 and 61; Lines 258 and 259). These studies use viral labeling to visualize the cells of interest during long-term (days-long) migration imaging without seemingly affecting organoid health, again as opposed to what's been suggested (Line 66). Please revise to reflect more accurately what's been achievable by the current state of the art. This surely does not take away from the novelty of the 3P/THG method but better clarifies the limitations being addressed. Please see references in Reviewer 3's comments below.

2) A major issue in the field of stem cell modeling of disease is how well the model recapitulates the disease. Please comment on how the phenotypes (less migration) in organoids inform our understanding of RTT. How does phenotype relate to patient neuropathology?

3) Is the RTT migration phenotype caused solely by the more tortuous VZ region they need to navigate? Once they reach THG-high regions (cortical plate), does the phenotype recover? If not, does this indicate additional cell-intrinsic deficits?

4) Please tone down the narrative in the introduction in which the dichotomy between intact 3D and sliced 2D structures is highlighted. While these are certainly different, both are still 3D and will benefit from deep tissue imaging techniques.

5) For clarity of the manuscript to a wide audience, please include some discussion contrasting THG with standard three-photon excitation.

6) An expanded discussion could serve readers well to envision the true potential of the 3P/THG system. Do the authors envision additional features can be extracted in older (e.g. myelinated axons) or novel (e.g. organoids with endothelia) organoids using their models? Is THG-coupled calcium imaging possible, for example where one can couple VZ calcium dynamics, such as waves (Weissman et al., 2004), to neuronal dynamics? Can other disorders that cause structural abnormalities like cancers, tuberous sclerosis, or heterotopias be modeled using organoids and differentiated from one another using the 3P/THG signal?

7) On L343 "incubation in index matching solution improved the imaging depth" does not seem to be described in the Methods, please describe. Please also clarify whether this was quantified through attenuation length, maximum depth at which cells could be observed, or another measure.

8) Some explanations are proposed for the result that WT and MT organoids show a different attenuation length. Were any measurements made to validate these explanations? The stated structural differences would appear to be measureable from 3D organoid stacks, and thus may provide support to the conjecture.

9) To better understand how the two different RTT iPSC isogenic pairs are measured and presented throughout the results, please use different colors or shapes to indicate data from the two lines so that it is clear that two lines were assessed and the data combined. Please also include more information in the results or legends for clarification of the data being presented (different lines, differences between lines, number of experimental batches, etc).

10) Please report actual p values, degrees of freedom (when relevant) and n for each t-test, and define the error bars (SEM, 90% CI, etc).

11) In Figure 6, it looks like the data is pooled from organoids that were imaged for different periods of time. If so, it is misleading to report statistical differences between WT and MT groups. Please clarify.

Note: Several validation studies were suggested in the reviews. The translation of THG signal to its cellular correlates is important and likely to present ongoing issues if this becomes a more commonplace imaging practice. However, given the strength of the technical advance and that this is the first proof-of-principle demonstration, we are not requiring further validation experiments at this time. If you already have data to address the issues, we highly encourage you to incorporate them into your revised manuscript.

*Reviewer #1 (Recommendations for the authors):*

Two major issues remain:

Throughout the results, it is not clear how the two different RTT iPSC isogenic pairs are measured and presented. Although some information is in Supplementary figures, the data would be better presented with the two lines separated by color/shape to make it clear that two lines were assessed and the data combined. Information in the figure legend or results should provide additional clarification. Were all data from different lines combined? Are there differences between lines? How many experimental batches were done?

A major issue in the field of stem cell modeling of disease is how well the model recapitulates the disease. The authors should comment on how the phenotypes (less migration) in organoids inform our understanding of RTT. How does phenotype relate to patient neuropathology?

*Reviewer #2 (Recommendations for the authors):*

I think the Intro slightly over-eggs some things in making the justification for the work. Firstly it is noted that sectioning induces unwanted tissue damage and cell loss. Yes, but this neglects that sectioning is intrinsically part of some cerebral organoid protocols, in order to mature them: as increasing numbers of neurons are produced, they are progressively pushed inside and undergo necrosis. The way of getting around this (and maturing 3D organoid structures over much longer timeframes) has been to slice the organoids and then maintain organotypic organoid slice cultures over long time periods. Thus the dichotomy made between intact 3D structures and sliced 2D structures in prior work is not really valid – there is much ongoing work on engineered cerebral organoids that occupy a middle ground.

For clarity of the manuscript to a wide audience, I think it would help a lot to contrast THG with standard three-photon excitation. Some readers may be confused (and not all 3P is THG). Several sentences around lines 68-69 might suffice. Citations [41-43] are made about THG in the brain, but really only [42] is about THG, the others relate to 3PEF.

Figure 2 – scale bars are missing lengths.

Figure 3 – You use both attenuation length and extinction length – are you referring to the same thing? Perhaps keep terminology consistent.

Figure 3 – please report the p-value and n for the t-test, it is not acceptable in this day and age to just say p<0.05. (Actually, I think it would be more useful to report effect size.) In the main text also: L219 is that an S.E.M. or 90% confidence interval? L220 state the test, n, df (if relevant), and actually produced a p-value. Given that from the figure, n=4, I'm not convinced that this is sufficiently powered, it seems to be in the danger zone, so should be reported and described carefully. Statistical reporting from L229 seems much better.

L260 what is DIV? Also, do you mean 1 frame being captured every 20 minutes or a sequence of frames?

Figure 6 – I am slightly puzzled by the displacement statistics shown here. It looks like the data pooled is from organoids that are imaged for different periods of time, in which case it is misleading to report statistical differences between WT and MT groups. I imagine the "straightness" measure would also be biased by the inclusion of samples recorded from different imaging durations.

L343 "incubation in index matching solution improved the imaging depth". I could not find this in the Methods, can you please detail? Also, did you quantify this through attenuation length, maximum depth at which cells could be observed, or another measure?

L355 you propose some explanations for the result that WT and MT organoids show a different attenuation length. Did you make any measurements to validate these explanations? The stated structural differences would appear to be measurable from 3D organoid stacks, and thus may provide support to the conjecture.

L413 exits rather than exists.

*Reviewer #3 (Recommendations for the authors):*

I have three broad critiques in regard to the cited literature, validation experiments, and additional applications:

1) Prior literature: Authors should revise the text to better reflect the current approaches in organoid imaging. There is a significant body of literature describing migration imaging using brain organoid models (e.g. Birey et al., 2017; Bagley et al., 2017; Xiang et al., 2017; Klaus et al., 2019; Bajaj et al., 2021; Birey et al., 2022), including ones that use intact (not sliced) assembloid preparations not cited in the paper (Xiang et al., 2017; Birey et al., 2017, 2022), as opposed to what's been suggested by the authors (Lines 60 and 61; Lines 258 and 259). These studies use viral labeling to visualize the cells of interest during long-term (days-long) migration imaging without seemingly affecting organoid health, again as opposed to what's been suggested (Line 66). Authors should revise to reflect more accurately what's been achievable by the current state of the art. This surely does not take away from the novelty of the 3P/THG method but better clarifies the limitations being addressed.

2) Validation experiments: In early characterization experiments, authors convincingly show that THG signal is higher in regions populated with neurons compared to denser VZ-like regions populated with progenitors. As it is important to translate the THG single as accurately as possible, similar validation experiments are needed to better link THG dynamics to specific cellular events in the following phenotyping experiments. For example, the authors show a higher surface area and about a 2-fold increase in the number of ventricles in RTT organoids. An optical clearing approach for the SA quantification and/or a simple cryosection quantification for the number of VZ-like zones should be able to reveal a change of this magnitude. Authors electroporate a CAG-driven plasmid to visualize cells that they have previously reported, albeit in three samples, to almost exclusively (~95%) capture neurons (Delepine et al., 2021). As the rationale as to why a generic promoter shows lineage selectivity is unclear and as the imaging field of interest also includes other cell types that migrate in a similar fashion (i.e. radial glia that go inter-kinetic nuclear migration), a parallel labeling strategy, e.g. lineage-specific viral vectors or expression vectors with different promoters, to further delineate the specific lineage captured by the THG signal would be informative. For example, Figure 3A and B show VZ-like regions with high TGH signal within their cores, which are unlikely to be neurons. This could resolve questions of this nature.

3) Is the RTT migration phenotype caused solely by the more tortuous VZ region they need to navigate? Once they reach THG-high regions (cortical plate), does the phenotype recover? If not, does this indicate additional cell-intrinsic deficits?

4) An expanded discussion could serve the reader well to envision the true potential of the 3P/THG system. Do the authors envision additional features can be extracted in older (e.g. myelinated axons) or novel (e.g. organoids with endothelia) organoids using their models? Is THG-coupled calcium imaging possible, for example where one can couple VZ calcium dynamics, such as waves (Weissman et al., 2004), to neuronal dynamics? Can other disorders that cause structural abnormalities like cancers, tuberous sclerosis, or heterotopias be modeled using organoids and differentiated from one another using the 3P/THG signal?

References:

Bagley, J.A., Reumann, D., Bian, S., Lévi-Strauss, J., and Knoblich, J.A. (2017). Fused cerebral organoids model interactions between brain regions. Nat Methods 14, 743-751. https://doi.org/10.1038/nmeth.4304.

Bajaj, S., Bagley, J.A., Sommer, C., Vertesy, A., Nagumo Wong, S., Krenn, V., Lévi-Strauss, J., and Knoblich, J.A. (2021). Neurotransmitter signaling regulates distinct phases of multimodal human interneuron migration. EMBO J 40, e108714. https://doi.org/10.15252/embj.2021108714.

Birey, F., Andersen, J., Makinson, C.D., Islam, S., Wei, W., Huber, N., Fan, H.C., Metzler, K.R.C., Panagiotakos, G., Thom, N., et al. (2017). Assembly of functionally integrated human forebrain spheroids. Nature 545, 54-59. https://doi.org/10.1038/nature22330.

Birey, F., Li, M.-Y., Gordon, A., Thete, M.V., Valencia, A.M., Revah, O., Paşca, A.M., Geschwind, D.H., and Paşca, S.P. (2022). Dissecting the molecular basis of human interneuron migration in forebrain assembloids from Timothy syndrome. Cell Stem Cell https://doi.org/10.1016/j.stem.2021.11.011.

Delepine, C., Pham, V.A., Tsang, H.W.S., and Sur, M. (2021). GSK3ß inhibitor CHIR 99021 modulates cerebral organoid development through dose-dependent regulation of apoptosis, proliferation, differentiation and migration. PLOS ONE 16, e0251173. https://doi.org/10.1371/journal.pone.0251173.

Klaus, J., Kanton, S., Kyrousi, C., Ayo-Martin, A.C., Di Giaimo, R., Riesenberg, S., O'Neill, A.C., Camp, J.G., Tocco, C., Santel, M., et al. (2019). Altered neuronal migratory trajectories in human cerebral organoids derived from individuals with neuronal heterotopia. Nat Med 25, 561-568. https://doi.org/10.1038/s41591-019-0371-0.

Weissman, T.A., Riquelme, P.A., Ivic, L., Flint, A.C., and Kriegstein, A.R. (2004). Calcium Waves Propagate through Radial Glial Cells and Modulate Proliferation in the Developing Neocortex. Neuron 43, 647-661. https://doi.org/10.1016/j.neuron.2004.08.015.

Xiang, Y., Tanaka, Y., Patterson, B., Kang, Y.-J., Govindaiah, G., Roselaar, N., Cakir, B., Kim, K.-Y., Lombroso, A.P., Hwang, S.-M., et al. (2017). Fusion of Regionally Specified hPSC-Derived Organoids Models Human Brain Development and Interneuron Migration. Cell Stem Cell 21, 383-398.e7. https://doi.org/10.1016/j.stem.2017.07.007.

---

## [Author Response]

Essential revisions:1) The prior literature in this area of research is not put in perspective well to reflect the current approaches in organoid imaging. There is a significant body of literature describing migration imaging using brain organoid models, including ones that use intact (not sliced) assembloid preparations not cited in the paper, as opposed to what's been suggested by the authors (Lines 60 and 61; Lines 258 and 259). These studies use viral labeling to visualize the cells of interest during long-term (days-long) migration imaging without seemingly affecting organoid health, again as opposed to what's been suggested (Line 66). Please revise to reflect more accurately what's been achievable by the current state of the art. This surely does not take away from the novelty of the 3P/THG method but better clarifies the limitations being addressed. Please see references in Reviewer 3's comments below.

We thank the reviewer for this comment, and have now included the suggested references in the main text of the revised manuscript (Lines 64-66). As a small point, while live cell imaging has been done in intact organoids with viral labeling, it is not clear that a viral strategy is the best for long-term imaging and preserving organoid health. In our recent paper performed in awake mouse brain (Yildirim et al., 2019), we found laser power regimes which have adverse (but not visible) effects on the physiological responses of virally labeled neurons, in addition to the higher laser power regime which can induce visible damage and optical breakdown. We have explained this briefly (Lines 69-70) as a further justification for using label-free approaches, importantly in conjunction with high-resolution deep-tissue imaging.

2) A major issue in the field of stem cell modeling of disease is how well the model recapitulates the disease. Please comment on how the phenotypes (less migration) in organoids inform our understanding of RTT. How does phenotype relate to patient neuropathology?

The vast majority of the literature has focused on the role of MeCP2 in postnatal brain maturation and function, mainly in mouse models, and relatively little is known about its effects in prenatal brain development. Structural deficits described in postmortem RTT human brains include reduced cortical thickness, cell size and dendritic arborization (Bauman et al., 1995; Armstrong et al., 1995), and reduced cerebral volume in MR imaging of RTT patients (Carter et al., 2008). These deficits are paralleled by reductions in dendritic arborization, soma size and spine density described in RTT mouse models (Smrt et al., 2007; Fukuda et al., 2005; Kishi and Macklis, 2004; Shahbazian et al., 2002). This is now mentioned in the Introduction (Lines 48-52).

Our previous work demonstrated impaired proliferation of the progenitor pool and delayed maturation and presumed migration of neurons (Mellios et al., 2018), consistent with the human postmortem deficits. However, as we point out, the use of fixed tissues did not allow cell tracking and analysis of parameters such as speed and trajectory of neuronal displacement; thus the dynamics and mechanisms of the presumptive migration deficit remain to be characterized. Our present findings directly demonstrate deficits in migration at very early stages of prenatal development which, together with deficits mentioned above and described as ‘curtailment of development’ (Bauman et al., 1995), provide insight into early events which underlie the structural phenotypes of cortical development in RTT. This is now noted in the Discussion (Lines 427-430).

3) Is the RTT migration phenotype caused solely by the more tortuous VZ region they need to navigate? Once they reach THG-high regions (cortical plate), does the phenotype recover? If not, does this indicate additional cell-intrinsic deficits?

We are not certain whether the RTT migration phenotype caused solely by more tortuous VZ region since we do not have long-term (potentially spanning several months) imaging data to reveal how cells migrate in the cortical plate. However, we have two working hypothesis that we would like to test. In the first hypothesis, the migration phenotype recovers when cells start to form the cortical plate since their differentiation into different neuronal cell types is fairly complete. We can address this by labeling neurons in the cortical plate with different cell-specific markers in addition to analyzing their migration patterns. In the second hypothesis, migration phenotypes especially for inhibitory neurons may continue while cortical layers are being formed. In this case, delayed neurodevelopment may cause disproportionately small and layered brain regions as well as disruption of excitation and inhibition (EI) balance. Although rare in the clinic, it is reported that some male babies born with a mutant MECP2 allele display disproportionately small frontal and temporal lobes characteristic of a prenatal pathogenesis (Schule et al., 2008). Recent studies have also demonstrated cell-autonomous deficits in GABA signaling, including that GABA is less hyperpolarizing in MeCP2-deficient mice (Banerjee et al., 2016) and in iPSC derived neurons generated from RTT patients (Tang et al., 2016).

4) Please tone down the narrative in the introduction in which the dichotomy between intact 3D and sliced 2D structures is highlighted. While these are certainly different, both are still 3D and will benefit from deep tissue imaging techniques.

We have removed the narrative in the introduction about dichotomy between intact 3D and sliced 2D structures. We now focus on challenges in 3D imaging of intact organoids and how to deal with them with label-free 3-photon imaging (Lines 64-73).

5) For clarity of the manuscript to a wide audience, please include some discussion contrasting THG with standard three-photon excitation.

We have expanded on our earlier description of three-photon fluorescence microscopy and THG microscopy in the Introduction. As reviewers suggested, we have now added more explanation about these microscopy techniques (Lines 79-96).

Both standard three-photon fluorescence microscopy and THG microscopy occur based on a three-photon interaction between ultrashort pulses and tissues. Three-photon fluorescence microscopy has been recently used to perform functional and structural brain imaging in anesthetized and awake mice (Ouzounov et al., 2017, Yildirim et al., 2019). These studies are based on utilizing a green (GCaMP) genetically engineered calcium indicator (exogenous fluorophores) with their corresponding excitation wavelengths (1300 nm) which provide peak absorption cross-sections for this indicator. In these three-photon fluorescence microscopy studies, three photons with enough peak power at these excitation wavelengths excite electrons from ground state to excitation states of this indicator. Then, these electrons release their energy while they return to their ground states. During this relaxation, they release emitted photons which have a large 1/e^2^ bandwidth of fluorescence emission (70 nm for GcaMP6s/f). Therefore, they require an exogenous label with large bandwidth of emission spectrum to perform three-photon fluorescence structural or functional brain imaging. On the other hand, THG microscopy does not need to have any label to perform structural three-photon brain imaging. In addition, three photons with enough peak power at the excitation wavelengths excite electrons from ground state to virtual state so that these electrons do not lose any energy when they come back to the ground state. Therefore, they release emitted photons which have a small 1/e^2^ bandwidth of emission (~20 nm), and the emission spectrum of THG imaging happens at exactly 1/3 of the excitation wavelength.

Overall, three-photon fluorescence microscopy is valuable for performing structural and functional brain imaging with fluorescent dyes for which the excitation wavelengths are limited by the peak absorption cross-sections of these fluorophores. However, three-photon fluorescence imaging is prone to phototoxicity and photobleaching particularly with high peak intensity pulses (Yildirim et al., 2019). In contrast, THG microscopy provides label-free structural brain imaging (Yildirim et al., 2020) without problems of phototoxicity and photobleaching – which are its strengths for long-term live cell imaging of cerebral organoids.

6) An expanded discussion could serve readers well to envision the true potential of the 3P/THG system. Do the authors envision additional features can be extracted in older (e.g. myelinated axons) or novel (e.g. organoids with endothelia) organoids using their models? Is THG-coupled calcium imaging possible, for example where one can couple VZ calcium dynamics, such as waves (Weissman et al., 2004), to neuronal dynamics? Can other disorders that cause structural abnormalities like cancers, tuberous sclerosis, or heterotopias be odelled using organoids and differentiated from one another using the 3P/THG signal?

At the reviewers’ suggestion, we have added a paragraph (Lines 436-444) to discuss possible applications of our label-free imaging system coupled with three-photon fluorescence imaging for different disorders. First, it is feasible to combine label free THG imaging and three-photon calcium imaging as shown in our recent paper (Yildirim et al., 2019). Therefore, we can also capture phenotypes in calcium activity in neurons in addition to their migration deficits in cerebral organoids. In addition, we can also capture structural phenotypes in myelinated axons and endothelial cells (microvasculature). We showed recently (Yildirim et al., 2020) that one can image calcium dynamics, myelinated axons, as well as microvasculature (three-color imaging) in the cortex and in the white matter of primary and higher visual areas of awake mice. We believe that imaging myelinated axons as well as endothelial cells can be used for studying brain disorders such as Multiple Sclerosis and Alzheimer’s Disease since there is a significant loss of myelin as well as reduction of blood flow in these diseases, respectively. Finally, it is possible to perform structural, and functional three-photon and THG imaging for revealing structural and functional phenotypes for cancer applications such as glioblastoma (Bakker et al., 2022).

7) On L343 "incubation in index matching solution improved the imaging depth" does not seem to be described in the Methods, please describe. Please also clarify whether this was quantified through attenuation length, maximum depth at which cells could be observed, or another measure.

We thank the reviewers for highlighting the effect of index matching solution on the imaging depth. We did not need to apply any index matching solution for 60% of the WT and MT organoids that we used in this study. For the remaining organoids, index matching gel improved the imaging depth up to about 50% as we quantified Supp. Figure 19 in the revised manuscript. We described the incubation of cerebral organoids in index-matching solution in the Methods (Lines 597-600).

8) Some explanations are proposed for the result that WT and MT organoids show a different attenuation length. Were any measurements made to validate these explanations? The stated structural differences would appear to be measureable from 3D organoid stacks, and thus may provide support to the conjecture.

We explained the differences in extinction lengths of WT and MT organoids based on the results that we presented in Figure 3C. We suggest that higher surface area with lower radial thickness in MT organoids resulted in irregular shapes of ventricular zones. In addition, number of ventricular zones in MT organoids is significantly higher than that of WT organoids. Therefore, higher number of irregular shaped ventricular zones may induce more scattering in MT organoids which results in their lower extinction lengths. We have added this additional explanation in the Discussion (lines 390-394).

9) To better understand how the two different RTT iPSC isogenic pairs are measured and presented throughout the results, please use different colors or shapes to indicate data from the two lines so that it is clear that two lines were assessed and the data combined. Please also include more information in the results or legends for clarification of the data being presented (different lines, differences between lines, number of experimental batches, etc).

We have used different colors (blue for Line 1 and green for Line 2) for each line in the Supplementary Figures 9-12, 14, 16-17. Also, we described each line that we used in the figure captions (Figures2-6, Supp. Figures 6-8,13, 15, 18-21).

10) Please report actual p values, degrees of freedom (when relevant) and n for each t-test, and define the error bars (SEM, 90% CI, etc).

We have reported actual p values, and n for each t-test and defined the error bars (SEM, or 90% CI) in each figure caption and in the main text (lines 250, 260, 262, 266, 268).

11) In Figure 6, it looks like the data is pooled from organoids that were imaged for different periods of time. If so, it is misleading to report statistical differences between WT and MT groups. Please clarify.

As reviewers pointed out, we agree that pooling the data from the cells that are not imaged for 12 hours is not reasonable for reporting statistical differences especially in displacement values of cells. These cells generally disappeared from the field of view before finishing 12 hours of imaging, therefore their values are limited to shorter time durations. We removed the data from these cells and updated figures 5 and 6 and supplementary figures 14,16-17 in the revised manuscript. Also, we updated the corresponding figure captions and reported values in the manuscript (lines 300-301, 304, 306, 314-315, 318-319).

Note: Several validation studies were suggested in the reviews. The translation of THG signal to its cellular correlates is important and likely to present ongoing issues if this becomes a more commonplace imaging practice. However, given the strength of the technical advance and that this is the first proof-of-principle demonstration, we are not requiring further validation experiments at this time. If you already have data to address the issues, we highly encourage you to incorporate them into your revised manuscript.

We thank the reviewers again for appreciating the technical advance of our paper, and we agree that it is the first proof-of-principle of THG imaging for intact cerebral organoids with application to Rett syndrome.

Reviewer #1 (Recommendations for the authors):Two major issues remain:Throughout the results, it is not clear how the two different RTT iPSC isogenic pairs are measured and presented. Although some information is in Supplementary figures, the data would be better presented with the two lines separated by color/shape to make it clear that two lines were assessed and the data combined. Information in the figure legend or results should provide additional clarification. Were all data from different lines combined? Are there differences between lines? How many experimental batches were done?

We explained whether we combined the data from both lines or not in each figure in the corresponding figure captions (Figures 2- 6 and Supp. Figures 5-8, 13, 15, 18, 20).

In terms of structural parameters (volume, surface area, volume per area and number of cerebral organoids), we observed no differences between WT and MT organoids of both lines. For example, MT organoids have larger surface area, lower volume/area thickness, and higher number of ventricular regions than those in WT organoids independent of the lines. When we focus on individual parameters, we can see some slight differences in absolute values of these parameters in MT and WT organoids in each line. For example, MT and WT organoids in Line 2 have higher number of ventricular regions as well as higher surface area compared to those in Line 1. Whereas, MT and WT organoids in Line 1 have higher volume/area thickness compared that of Line 2. Finally, WT organoids in Line 2 have higher ventricular volume compared to that in Line 1 whereas MT organoids in Line 1 have higher ventricular volume than that of Line 2.

In terms of migration parameters (overall displacement, average speed, and straightness), we found no differences between WT and MT organoids of both lines. When we focus on individual parameters, we found that both WT and MT organoids in Line 2 have higher average speed, straightness and longer displacement than those in Line 1.

A major issue in the field of stem cell modeling of disease is how well the model recapitulates the disease. The authors should comment on how the phenotypes (less migration) in organoids inform our understanding of RTT. How does phenotype relate to patient neuropathology?

We have added information about the relevance of migration deficits to neuropathology of RTT in the Introduction and Discussion, as described in item 2 of Essential Revisions.

Reviewer #2 (Recommendations for the authors):I think the Intro slightly over-eggs some things in making the justification for the work. Firstly it is noted that sectioning induces unwanted tissue damage and cell loss. Yes, but this neglects that sectioning is intrinsically part of some cerebral organoid protocols, in order to mature them: as increasing numbers of neurons are produced, they are progressively pushed inside and undergo necrosis. The way of getting around this (and maturing 3D organoid structures over much longer timeframes) has been to slice the organoids and then maintain organotypic organoid slice cultures over long time periods. Thus the dichotomy made between intact 3D structures and sliced 2D structures in prior work is not really valid – there is much ongoing work on engineered cerebral organoids that occupy a middle ground.

We changed the wording about dichotomy between 2d and 3d preparations of cerebral organoids in the Introduction and Discussion. Please see item 4 in Essential Revisions.

For clarity of the manuscript to a wide audience, I think it would help a lot to contrast THG with standard three-photon excitation. Some readers may be confused (and not all 3P is THG). Several sentences around lines 68-69 might suffice. Citations [41-43] are made about THG in the brain, but really only [42] is about THG, the others relate to 3PEF.

We have added a section to contrast THG and standard three-photon fluorescence imaging in the Introduction. Please refer to the item 5 in the Essential Revisions.

Figure 2 – scale bars are missing lengths.

We provided the lengths of all scale bars in Figure 2.

Figure 3 – You use both attenuation length and extinction length – are you referring to the same thing? Perhaps keep terminology consistent.

We termed all the lengths (combined scattering and absorption) as extinction lengths in the main text.

Figure 3 – please report the p-value and n for the t-test, it is not acceptable in this day and age to just say p<0.05. (Actually, I think it would be more useful to report effect size.) In the main text also: L219 is that an S.E.M. or 90% confidence interval? L220 state the test, n, df (if relevant), and actually produced a p-value. Given that from the figure, n=4, I'm not convinced that this is sufficiently powered, it seems to be in the danger zone, so should be reported and described carefully. Statistical reporting from L229 seems much better.

For the figure 3, we added two more samples for both WT and MT to increase the statistical power and updated Figure 3E.

L260 what is DIV? Also, do you mean 1 frame being captured every 20 minutes or a sequence of frames?

DIV means days in vitro. We captured 5 frames on the same plane and moved to another z plane which is 5 µm away from the previous plane. We continued getting these z-stack images of 50-100 different planes to make a volume. This volumetric imaging took approximately 3-6 minutes. Then, we repeated this volumetric acquisition at every 20 minutes. We updated the text for explaining the DIV term and speed of data acquisition (Lines 293,478-480).

Figure 6 – I am slightly puzzled by the displacement statistics shown here. It looks like the data pooled is from organoids that are imaged for different periods of time, in which case it is misleading to report statistical differences between WT and MT groups. I imagine the "straightness" measure would also be biased by the inclusion of samples recorded from different imaging durations.

We agree with the reviewer that comparing the displacement and straightness values of the cells which disappeared from the field of view before finalizing the overall imaging duration of 12 hours may not be reasonable. Therefore, we removed these data and updated Figure 6 as well as Figure 5. Please see item 11 in the Essential Revisions.

L343 "incubation in index matching solution improved the imaging depth". I could not find this in the Methods, can you please detail? Also, did you quantify this through attenuation length, maximum depth at which cells could be observed, or another measure?

We would like to thank the reviewer for highlighting the effect of index matching solution on the imaging depth. We did not need to apply any index matching solution for 60% of the WT and MT organoids that we used in this study. For the remaining part of the organoids, index matching gel improved the imaging depth maximum 50% as we quantified in Supp. Figure 19 in the revised manuscript. Please see item 7 above in Essential Revisions.

L355 you propose some explanations for the result that WT and MT organoids show a different attenuation length. Did you make any measurements to validate these explanations? The stated structural differences would appear to be measurable from 3D organoid stacks, and thus may provide support to the conjecture.

We explained the differences in extinction lengths of WT and MT organoids based on the results that we presented in Figure 3C. We suggest that higher surface area with lower radial thickness in MT organoids resulted in irregular shapes of ventricular zones. In addition, number of ventricular zones in MT organoids is significantly higher than that of WT organoids. Therefore, higher number of irregular shaped ventricular zones may induce more scattering in MT organoids which results in their lower extinction lengths. We added this additional explanation in the Discussion part (lines 390-394, also see item 8 in Essential Revisions).

L413 exits rather than exists.

We corrected this typo (Line 461)

Reviewer #3 (Recommendations for the authors):I have three broad critiques in regard to the cited literature, validation experiments, and additional applications:1) Prior literature: Authors should revise the text to better reflect the current approaches in organoid imaging. There is a significant body of literature describing migration imaging using brain organoid models (e.g. Birey et al., 2017; Bagley et al., 2017; Xiang et al., 2017; Klaus et al., 2019; Bajaj et al., 2021; Birey et al., 2022), including ones that use intact (not sliced) assembloid preparations not cited in the paper (Xiang et al., 2017; Birey et al., 2017, 2022), as opposed to what's been suggested by the authors (Lines 60 and 61; Lines 258 and 259). These studies use viral labeling to visualize the cells of interest during long-term (days-long) migration imaging without seemingly affecting organoid health, again as opposed to what's been suggested (Line 66). Authors should revise to reflect more accurately what's been achievable by the current state of the art. This surely does not take away from the novelty of the 3P/THG method but better clarifies the limitations being addressed.

We updated the Introduction by adding the references that Reviewer 3 suggested. Please refer to item 1 above in the Essential Revisions.

2) Validation experiments: In early characterization experiments, authors convincingly show that THG signal is higher in regions populated with neurons compared to denser VZ-like regions populated with progenitors. As it is important to translate the THG single as accurately as possible, similar validation experiments are needed to better link THG dynamics to specific cellular events in the following phenotyping experiments. For example, the authors show a higher surface area and about a 2-fold increase in the number of ventricles in RTT organoids. An optical clearing approach for the SA quantification and/or a simple cryosection quantification for the number of VZ-like zones should be able to reveal a change of this magnitude. Authors electroporate a CAG-driven plasmid to visualize cells that they have previously reported, albeit in three samples, to almost exclusively (~95%) capture neurons (Delepine et al., 2021). As the rationale as to why a generic promoter shows lineage selectivity is unclear and as the imaging field of interest also includes other cell types that migrate in a similar fashion (i.e. radial glia that go inter-kinetic nuclear migration), a parallel labeling strategy, e.g. lineage-specific viral vectors or expression vectors with different promoters, to further delineate the specific lineage captured by the THG signal would be informative. For example, Figure 3A and B show VZ-like regions with high TGH signal within their cores, which are unlikely to be neurons. This could resolve questions of this nature.

We would like to thank the reviewer for the suggestions. In future follow-up studies, multi-color multiphoton imaging (including THG) can be performed to delineate further the cell-type and/or lineage-type information we can obtain for migration experiments in details.

3) Is the RTT migration phenotype caused solely by the more tortuous VZ region they need to navigate? Once they reach THG-high regions (cortical plate), does the phenotype recover? If not, does this indicate additional cell-intrinsic deficits?

Please see item 3 in the Essential Revisions.

4) An expanded discussion could serve the reader well to envision the true potential of the 3P/THG system. Do the authors envision additional features can be extracted in older (e.g. myelinated axons) or novel (e.g. organoids with endothelia) organoids using their models? Is THG-coupled calcium imaging possible, for example where one can couple VZ calcium dynamics, such as waves (Weissman et al., 2004), to neuronal dynamics? Can other disorders that cause structural abnormalities like cancers, tuberous sclerosis, or heterotopias be modeled using organoids and differentiated from one another using the 3P/THG signal?

We have added this at the end of Discussion. Please see item 6 in the Essential Revisions.